# Morphology of *Vavilovia formosa* (Steven) Fed. Nodules Induced by Different Rhizobia Strains

**DOI:** 10.3390/plants14243764

**Published:** 2025-12-10

**Authors:** Anna V. Tsyganova, Artemii P. Gorshkov, Anastasiia K. Kimeklis, Olga P. Onishchuk, Maxim G. Vorobiev, Evgeny E. Andronov, Viktor E. Tsyganov

**Affiliations:** 1All-Russia Research Institute for Agricultural Microbiology, Saint Petersburg 196608, Russia; a.gorshkov@arriam.ru (A.P.G.); akimeklis@arriam.ru (A.K.K.); o.onishuk@arriam.ru (O.P.O.); e.andronov@arriam.ru (E.E.A.); vetsyganov@arriam.ru (V.E.T.); 2Centre for Molecular and Cell Technologies, Saint Petersburg State University, Saint Petersburg 199034, Russia; vorobiev.maxim@spbu.ru

**Keywords:** *Vavilovia formosa* (Steven) Fed., nodule morphology, infection thread, symbiosome, bacteroid

## Abstract

The study of wild relatives of crop legumes offers an inexhaustible source of useful properties and microorganisms for agriculture. In this study, we morphologically examined vavilovia (*Vavilovia formosa* (Steven) Fed.) nodules induced by strains of *Rhizobium ruizarguesonis* RCAM1026, *Rhizobium leguminosarum* sv. *viciae* TOM, as well as *R. leguminosarum* sv. *viciae* strains Vaf-12 and Vaf-108, isolated from nodules of wild-type plants of *V. formosa*. The nodules induced by *R. leguminosarum* sv. *viciae* strain Vaf-12 maintained histological and ultrastructural organization typical for indeterminate nodules. Different ultrastructural abnormalities were revealed in nodules induced by the other strains. When using the *R. leguminosarum* sv. *viciae* strain Vaf-108, pinkish nodules formed, in which a senescence zone developed in four weeks after inoculation. Furthermore, small brownish pseudonodules were also formed. In the nodules induced by the *R. leguminosarum* sv. *viciae* strain TOM bacteroids rapidly degraded and were excluded from the cytoplasm into the vacuole. Nodules induced by the *R. ruizarguesonis* strain 1026 were characterized with excessive accumulation of starch grains in mature infected cells.

## 1. Introduction

Plant breeding is constantly faced with new challenges, such as increasing productivity per unit of sown area due to the rapidly growing population of the Earth, as well as adapting crops to a changing climate [1]. Legumes are considered crucial to the development of environmentally friendly agriculture due to their high protein content and low dependence on mineral nitrogen fertilizers, as legumes in symbiosis with rhizobia are able to fix atmospheric nitrogen [2,3]. That is why symbiotic nitrogen fixation is one of the most promising traits for breeding of legume crops.

The Fabeae is one of the richest tribes in Fabaceae Lindl. It is estimated to include over 300 species, whose taxonomic structure is still quite dynamic. Many species of the Fabeae are of extreme economic importance. These include garden pea (*Pisum sativum* L. synonym *Lathyrus oleraceus* Lam.), faba bean (*Vicia faba* L.), lentil (*Lens culinaris* Medik.), grass pea (*Lathyrus sativus* L.), and common vetch (*Vicia sativa* L.). All these species can be considered as multifunctional crops used for human consumption, animal feed, and numerous non-food purposes [4].

Until recently, the Fabeae was considered to consist of five genera. Two of them, *Lathyrus* L. and *Vicia* L., include more than 100 species each. Two more genera, *Lens* Mill. and *Pisum* L., have fewer species, and the fifth is a monotypic genus, *Vavilovia* Fed. Currently, pea and vavilovia species were included in the genus *Lathyrus* L., as *L. oleraceus* Lam. and *Lathyrus formosus* (Steven) Kenicer, accordingly [5]. However, the taxonomy of the genus *Lathyrus* is currently still a matter of debate [6].

Vavilovia is a perennial endemic to the mountains of Caucasus, Armenia, Anatolia, Iran and Iraq [7]. Its uniqueness as an object of study lies in its morphology and ecological traits [7]. It grows in a high-mountain regions in small populations, intersecting with the estimated center of origin of peas [8].

Vavilovia has recently attracted increasing interest due to its potential to provide legume researchers with valuable information on the past, present and future of the entire Fabeae [9]. Vavilovia shares the same chromosome number with close crop relatives such as pea, vetch and grass pea, opening up the possibility of hybridization [4]. In addition, the rhizobial symbionts of vavilovia were isolated and described recently [10,11,12]. However, the morphology of vavilovia nodules, which belong to the indeterminate type, has not been studied before.

Typically, nodules of indeterminate type are characterized by a persistent meristem, infection zone, interzone II–III, nitrogen fixation zone, and senescence zone [13]. Infection threads—tubular structures bounded by the cell wall—penetrate the infection zone [14]. The infection thread forms unwalled outgrowths, called infection droplets [13]. Bacteria are released from infection droplets into the plant cell cytoplasm, remaining surrounded by a plant-derived membrane containing bacterial components, called the peribacteroid (symbiosome) membrane [15]. The released bacteria undergo differentiation, transforming into a specialized form—a bacteroid. The bacteroid and the peribacteroid space, surrounded by the symbiosome membrane, form an organelle-like symbiosome [16]. Hosting of plant cells by symbiosomes requires their differentiation, accompanied by a significant increase in their size. As a result, infected cells achieve full functionality in the nitrogen fixation zone. Some cells remain uninfected. The functioning of the nodule ends with the formation of a senescence zone at its base, which gradually spreads towards the apex of the nodule [17].

In this study, the histological and ultrastructural traits of vavilovia nodules induced by different strains of rhizobia, with particular emphasis on the differentiation of bacteroids, were analyzed.

## 2. Results

Pink (Figure 1B–D) or pinkish (Figure 1E) nodules were formed on plants of vavilovia inoculated with different strains of rhizobia (Figure 1A). However, inoculation with *R. leguminosarum* sv. *viciae* Vaf-108 strain also resulted in the formation of small, brownish nodules.

All strains induced nodules with typical for indeterminate nodules zonation (meristem, an infection zone, an interzone II–III, and a nitrogen fixation zone) (Appendix A). However, in pinkish nodules induced by *R. leguminosarum* sv. *viciae* Vaf-108 the senescence zone was also distinguishable, while brownish nodules were empty pseudonodules (Appendix A). Also, in the case of inoculation with *R. leguminosarum* sv. *viciae* strain TOM, individual degenerating cells were found at the base of the nodule (Appendix A).

The peripheral tissues were presented by nodule parenchyma, in which typical vascular bundles pass, a single layer of nodule endodermis with thickened walls and the cortex consisting of several layers of cells (Appendix A).

To study in detail the structure of nodules induced by four strains, an analysis of their histological and ultrastructural organization was carried out.

### 2.1. Structure of Nodules Induced by the Rhizobium leguminosarum sv. viciae Strain Vaf-12

The meristem consisted of small, tightly packed cells (Figure 2A) with cytoplasm filled with ribosomes, proplastids, mitochondria, and several small vacuoles (Figure 2B). The nuclei contained one large nucleolus and weakly condensed chromatin. Mitotic figures were observed (Figure 2A). The intercellular spaces were filled with a medium-density matrix, and numerous granular deposits were observed on the inner side of the cell wall (Figure 2B).

The cells in the infection zone increased in size (Figure 3A,B); infection threads had a fibrillar wall of medium thickness (Figure 3C,D). Bacteria in the infection thread were not surrounded by exopolysaccharide capsules and an electron-transparent ring was detected around the infection thread matrix (Figure 3C,D), which was also observed in the infection droplet (Figure 3E). The bacteria were released from infection droplets individually, and some of them were probably surrounded by an EPS capsule, while others were not (Figure 3E). Small juvenile bacteroids with a matrix of medium electron density were located at the cell periphery (Figure 3A,F). Uninfected cells in the infection zone contained a nucleus and a thin layer of cytoplasm (Figure 3A,B).

Infected cells in interzone II–III had dense dark cytoplasm filled with differentiating bacteroids (Figure 4A,B). At the periphery, infected cells were filled with starch grains (Figure 4A).

In the nitrogen fixation zone, the infected cells increased significantly in size (Figure 4E) and contained a large central vacuole. The nucleus with a small nucleolus and abundant dense chromatin regions was positioned close to the vacuole (Figure 4H). The infection thread wall in the nitrogen fixation zone became very thin, non-fibrillar, and the electron-transparent ring almost disappeared (Figure 4D). Several amorphous inclusions with increased electron density were observed on the plasma membrane or on the outer surface of the infection thread wall (Figure 4D). The exopolysaccharide capsule around the bacteria was also not detected (Figure 4D). Mature differentiated bacteroids had a pleiomorphic form (Figure 4E,F). Inclusions of various electron density were found in the bacteroids (Figure 4F,G). Numerous starch grains appeared in uninfected cells in the nitrogen fixation zone (Figure 4C,E).

These nodules lacked a senescence zone. Occasionally, mature infected cells were observed with large body, probably of lipid composition (Figure 4H). Rare individual degenerating bacteroids were observed in mature infected cells, distinguished by an electron-opaque matrix, uneven peripheral cytoplasm (Figure 4G).

Free ribosomes and numerous profiles of rough endoplasmic reticulum (Figure 3C), some of which were enlarged (Figure 4B), were detected in infected cells. Amyloplasts and mitochondria were numerous at three-cell junctions (Figure 4B). Furthermore, Golgi bodies and various vesicles were frequent in the cytoplasm (Figure 3F).

### 2.2. Structure of Nodules Induced by the Rhizobium leguminosarum sv. viciae Strain Vaf-108

Meristem cells were irregularly shaped, small in size, and had dense cytoplasm (Figure 5A) containing numerous free ribosomes, proplastids, mitochondria, enlarged endoplasmic reticulum cisterns, and lipid bodies (Figure 5B). Vacuoles in meristematic cells were numerous and small. Nuclei were often lobed, had one or two large nucleoli, and a few clumps of condensed chromatin. Mitoses were visible in meristematic cells (Figure 5A). The nodule parenchyma cells also accumulated lipid bodies in their cytoplasm (Figure 5B).

In the infection zone, the infected cells increased in size (Figure 6A,B). The nucleus contained a large nucleolus and was located in the center of the cell (Figure 6B). Infection threads with an amorphous wall of varying thickness and outgrowths were often found in the infected cells (Figure 6C). Numerous releases of bacteria into the cytoplasm from infection droplets were observed (Figure 6D). In infection threads and droplets, an electron-transparent ring around the matrix was found, while the bacteria lacked an exopolysaccharide capsule (Figure 6C,D). Small juvenile bacteroids were freely distributed at the cell periphery (Figure 6B,F). Uninfected cells were filled with starch grains (Figure 6B). The intercellular spaces at the junctions of three cells were filled with an electron-dense matrix (Figure 6E), which later cleared in the nitrogen fixation zone. Lipid bodies were often observed along the cell walls on the plasma membranes at the sites of penetration of infection threads (Figure 6C).

In the interzone II–III and nitrogen fixation zone, the infected cells increased significantly in size and contained a nucleus with nucleolus abutted to a large central vacuole (Figure 7A,C). The infection thread wall in the nitrogen fixation zone became thinner, but acquired a fibrillar structure and, as well as in the infection zone, a transparent ring was observed around the matrix (Figure 7D). As in the infection zone, lipid bodies were observed at the sites of penetration of infection threads (Figure 7B). Mature differentiated bacteroids had a pleiomorphic shape and inclusions of varying electron density (Figure 7B,D–F). Bacteroids with large granules of polyhydroxybutyrate (PHB) were often encountered (Figure 7E).

In some nodules induced by *R. leguminosarum* sv. *viciae* strain Vaf-108, small senescence zone was formed, while in the other only individual degenerating infected cells were observed (Figure 7G). In such cells, the degenerating bacteroids had an uneven surface, electron transparent rounded areas in the peripheral cytoplasm, and membrane coil in the cytoplasm (Figure 7F). Among degenerating bacteroids, fusion of symbiosome membranes was observed resulting in formation of multibacteroid symbiosomes (Figure 7H).

### 2.3. Structure of Nodules Induced by the Rhizobium leguminosarum sv. viciae Strain TOM

Meristematic cells were small, overally electron-dense (Figure 8A) and contained numerous vacuoles, plastids, some containing starch grains, a few mitochondria, large numbers of free ribosomes, and occasional endoplasmic reticulum profile (Figure 8B). The nuclei of meristematic cells contained one to three nucleoli and large amounts of condensed chromatin. In addition to nucleoli, various inclusions similar to Cajal bodies were determined (Figure 8B). The meristematic cells of the proximal layer usually had infection thread profiles growing from the infection zone (Figure 8B). Numerous lipid bodies were found in the cells of the nodule parenchyma (Figure 8C).

In infected cells of the infection zone, numerous vacuoles fused into single one (Figure 9A,B). However, abundant profiles of small vacuoles, presumably expansions of the endoplasmic reticulum cisterns, could still be distinguished in the cytoplasm (Figure 9B). The nucleus with a large nucleolus was positioned into the cytoplasm-rich part of the cell. Profiles of infection threads and infection droplets were observed in an infected cell, from which bacteria were released into the cytoplasm (Figure 9A). The infection threads had a thick fibrillar wall, and the bacteria lacked an exopolysaccharide capsule (Figure 9C). Infection droplets contained a large number of rhizobia surrounded by a matrix of medium electron density (Figure 9D). Bacterial cells were surrounded by a light circle of an exopolysaccharide capsule, which they lost upon release into the cytoplasm (Figure 9D). In the infection zone, intercellular spaces were filled with an electron-opaque matrix (Figure 9E). Small juvenile bacteroids had a cytoplasm of medium electron density with an increase in density at the edges (Figure 9E) and were distributed along the periphery of the cell. Sometimes, infected cells in which juvenile bacteroids gathered into multibacteroid symbiosomes were encountered in the infection zone (Figure 9F).

Infected cells from interzone II–III had starch grains at the periphery, and numerous round inclusions were detected in the vacuoles (Figure 10A). Electron microscopic examination revealed these inclusions to be abnormal bacteroids (Figure 10B). Before being released into the vacuole, these bacteroids significantly increased in size and acquired a spherical shape (Figure 10C). Then their cytoplasm unevenly condensed, and one or two electron-opaque inclusions, likely nucleoids, were detected within (Figure 10D). Multibacteroid symbiosomes were also detected in infected cells (Figure 10B,C).

In the nitrogen fixation zone, the infected cells increased significantly in size (Figure 10E). The nucleus with a small nucleolus and abundant dense chromatin regions was positioned close to the vacuole. Infection threads in the nitrogen fixation zone had a thinner wall in which no fibrillar content was observed (Figure 10F). A transparent ring has been developed around the infection thread matrix, and the bacteria also lacked a capsule (Figure 10F). Bacteroids differentiated into mature nitrogen-fixing bacteroids of pleiomorphic form (Figure 10G).

The nodules lacked a senescence zone, but there were a few degenerating cells with destroyed cytoplasm (Figure 10H), degenerating bacteroids (Figure 10I), and starch grains that almost completely filled the cell space (Figure 10H).

### 2.4. Structure of Nodules Induced by the Rhizobium ruizarguesonis Strain RCAM1026

The meristem consisted of small cells with dark cytoplasm, in which numerous small vacuoles were determined (Appendix A). Among other organelles, proplastids, mitochondria, free ribosomes, and a few profiles of endoplasmic reticulum cisterns were observed in the meristematic cells (Appendix A). The nuclei were characterized by the presence of several nucleoli, and condensed chromatin (Appendix A). The intercellular spaces were filled with an electron-dense matrix; on the inner side of the cell wall adjacent to the plasma membrane, deposits in the form of electron-dense granules were observed (Appendix A).

The cells in the infection zone increased in size, and small vacuoles fused into one or two vacuoles (Appendix A). The nucleus contained a large nucleolus and was usually adjacent to the infection thread and droplet. Infection threads in the infection zone had a medium-thick wall with a pronounced fibrillar inner layer (Appendix A). In both infection threads and infection droplets, an electron-transparent ring around the matrix was observed, while the rhizobia lacked an exopolysaccharide capsule (Appendix A). Rhizobia released from infection droplets had opaque cytoplasm (Appendix A). Small juvenile bacteroids were located along the cell periphery and their cytoplasm was of medium electron density with an increase in density at the cytoplasm perimeter (Appendix A).

Instead of typical interzone II–III, polygonal cells lacking starch were found (Appendix A).

In the nitrogen fixation zone, the infected cells increased significantly in size, contained a centrally located vacuole and numerous amyloplasts (Appendix A). The nucleus with a small nucleolus and numerous abundant dense chromatin regions was positioned close to the vacuole (Appendix A). The infection thread wall in the nitrogen fixation zone became thinner and the fibrillar structure of this wall was not discernible (Appendix A). Mature differentiated bacteroids were pleiomorphic (Appendix A). Small mitochondria, endoplasmic reticulum and Golgi bodies were located in the cytoplasm around the bacteroids (Appendix A). Sometimes, individual degenerated bacteroids could be observed in mature infected cells, distinguished by an electron-denser, heterogenous cytoplasm (Appendix A). Uninfected cells were also filled with amyloplasts (Appendix A).

The nodules lacked a senescence zone; however, in their proximal portion they contained a few degenerating cells with pycnotic nuclei and lysed cytoplasm (Appendix A). In some degenerating cells, infection threads were sometimes observed, the matrix of which was of increased electron density with bacteria trapped within the walls (Appendix A). In such cells with condensed cytoplasm, major organelles such as the endoplasmic reticulum, Golgi bodies, and free ribosomes were virtually indistinguishable (Appendix A). Mitochondria and amyloplasts, like bacteroids, were surrounded by condensed cytoplasm. Bacteroids had normal morphology, but were located in large vacuole-like formations surrounded by a thin layer of compact cytoplasm in which other organelles were not detected (Appendix A).

## 3. Discussion

In the present study, the structure of vavilovia nodules induced by different rhizobial strains was analyzed in detail for the first time. All strains induced formation of indeterminate nodules, typical of a wide range of different legumes, such as pea [18,19], barrel medic (*Medicago truncatula* Gaertn.) [20,21], alfalfa (*Medicago sativa* L.) [22], bitter vetch (*Lathyrus linifolius*) [23], common vetch [24], and faba bean [25]. However, only nodules induced by *R. leguminosarum* sv. *viciae* strain Vaf-12 demonstrated the histological and ultrastructural organization typical of indeterminate nodules. Various ultrastructural abnormalities were detected in the nodules induced by other strains. In particular, the structure of nodules induced by strain RCAM1026 on vavilovia differed significantly from that of pea nodules formed after inoculation of plants with this strain [26].

All studied strains induced nodules in which typical meristematic cell organization was observed. Nodule meristem in pea—vavilovia’s close relative—consists of small isodiametric cells, which mitotically divide [19,27]. Ultrastructurally, meristematic cells are characterized by dense cytoplasm, large nuclei with one or more nucleoli, several small vacuoles, and proplastids and mitochondria freely located in the cytoplasm. Frequently dividing cells at the proximal part of the meristem are subject to invasion by infection threads [19]. However, in the present study, some features inconsistent with this typical picture were revealed. Lipid bodies were found in the meristematic cells of nodules induced by *R. leguminosarum* sv. *viciae* strain Vaf-108 (Figure 5B). Numerous lipid bodies have previously been shown in nodules of the arctic species *Oxytropis maydelliana* Trautv., *O. arctobia* Bunge, *Astragalus alpinus* L., and beach pea (*Lathyrus maritimus* (L.) Bigel.), which is presumably related to the growth conditions of the Arctic climate [28,29]. It is interesting to note that similar lipid bodies were also observed in this study in uninfected cells when inoculated with the *R. leguminosarum* sv. *viciae* strain TOM (Figure 8C).

Infection of nodule cells occurs through the release of bacteria from unwalled infection droplets, which form as lateral outgrowths on infection threads surrounded by the infection thread wall similar in structure and composition to plant cell wall [14,24]. When analyzing infection thread profiles in different nodule zones, it was found that infection threads in cells of the infection zone often had thickened cell walls with a distinct fibrillar layer (Figure 3C,D, Figure 9C and Appendix A). Subsequently, in cells of the nitrogen fixation zone, the infection threads walls were rather modified and became thinner and nonfibrillar (Figure 4D, Figure 10F and Appendix A). It is possible that differences in the structure of the wall of infection threads in the infection zone and the nitrogen fixation zone are associated with the intensive growth of infection threads in the infection zone and cessation of infection thread growth in the nitrogen fixation zone. Previously, in pea nodules, it was described that the wall of the infection thread is similar to the primary wall of a plant cell and does not have a pronounced fibrillar layer [19,30,31]. However, the presence of a fibrillar layer in the infection thread wall has been described in nodules of a symbiotically ineffective pea mutant RisFixV [32], in a case of partial reduction in *DOES NOT MAKE INFECTIONS 2* gene expression in *M. truncatula* nodules [33], and in pea nodules exposed to short-term aluminum treatment [34,35]. However, in nodules induced by *R. leguminosarum* sv. *viciae* strain Vaf-108, the preservation of the fibrillar layer in the cell walls of infection threads in the nitrogen fixation zone was observed (Figure 7D), while in the infection zone, the thickened walls of infection threads formed a variety of outgrowths (Figure 6C). Similar outgrowths of infection thread walls have been previously detected during pea treatment with fungicides [36] and aluminum [35], as well as in infection threads in pseudonodules of transgenic alfalfa plants carrying genes encoding pea seed lectin and inoculated with *R. leguminosarum* sv. *viciae* [37]. In vavilovia nodules, the modification of the infection thread wall with the formation of a fibrillar layer may be associated with the perennial lifestyle of this plant. However, we were unable to test this hypothesis in our study, as we examined four-week-old nodules.

A striking feature of the analyzed nodules was the absence of an exopolysaccharide capsule around bacteria in infection threads and droplets (Figure 3C,E, Figure 4D, Figure 6C,D, Figure 7D, Figure 8B, Figure 9C, Figure 10F, Appendix A). It well known that EPS are essential for the establishment of nitrogen-fixing symbiosis in legumes developing indeterminate nodules [38,39]. However, many studies demonstrated that EPS can be dispensable for a successful interaction, particularly in symbiotic pairs that form determinate nodules. In addition, it was discovered that in a symbiotic pair *Sinorhizobium fredii* HH103/*Glycyrrhiza uralensis* Fisch. ex DC. forming indeterminate nodules, the EPS does not play an important role in establishment of the interaction [40]. Despite the absence of an EPS capsule around the bacteria, the formation of an electron-transparent ring around the matrix of infection droplets and threads was observed (Figure 3C, Figure 4D, Figure 6C,D, Figure 7D, Figure 9D, Figure 10F and Appendix A). It is tempting to speculate that this ring may be associated with EPS, but this assumption requires further research.

At the same time, no visible abnormalities in the development of infection droplets were observed in the nodules induced by all four strains (Figure 3E, Figure 6D, Figure 9D and Appendix A). Bacteria released from infection droplets, individually surrounded by symbiosome membrane (Figure 3E, Figure 6D, Figure 9D and Appendix A), as well as mitochondria and plastids, shifted to the cell periphery (Figure 4B, Figure 6E, Figure 9E and Appendix A). Juvenile bacteroids in infected cells of the infection zone were virtually identical in size and cytoplasm density to bacteria in infection threads and droplets (Figure 3F, Figure 6F, Figure 9E and Appendix A).

In all analyzed variants of inoculation, during cell infection and transition from the meristem to the infection zone, the fusion of small vacuoles in infected cells into large ones, typical for indeterminate nodules was observed [19,41]; at the same time, the large nucleus with a prominent nucleolus was located at the center of the cell (Figure 3A,B, Figure 6B, Figure 9A and Appendix A). Unlike other strains, numerous expanded endoplasmic reticulum profiles were observed in nodules inoculated with *R. leguminosarum* bv. *viciae* strain TOM, giving the cytoplasm a “lace-like” appearance (Figure 9B).

In the nitrogen fixation zone, typically for indeterminate nodules [19,20,21,26,42,43,44], infected cells significantly enlarged and became filled with numerous symbiosomes (Figure 4E, Figure 7C, Figure 10E and Appendix A). In the case of inoculation with *R. ruizarguesonis* strain RCAM1026, numerous amyloplasts were deposited along the periphery of infected cells from the nitrogen fixation zone (Appendix A). Such starch accumulation is observed in ineffective nodules [45]. But for perennial plants such as beach pea, it has been shown that the presence of numerous starch granules in infected and uninfected cells can serve as a carbohydrate reserve and help to maintain high osmolarity of cells so that the dormant nodules do not freeze and they overcome winter conditions [46]. In the case of inoculation with *R. leguminosarum* sv. *viciae* strain Vaf-12, large lipid bodies with increased electron density were occasionally observed near the nucleus in some infected cells (Figure 4H). To our knowledge, the formation of such lipid bodies near the nucleus has not been previously described in infected nodule cells. Lipid body accumulation has previously been shown for arctic legumes, but they accumulated as small, numerous drops throughout the cytoplasm of both infected and uninfected cells [28,29].

Mature nitrogen-fixing bacteroids in nodules induced by all studied strains were individually enclosed by the symbiosome membrane, elongated compared to juvenile bacteroids, acquired diverse shapes, branched, and their cytoplasm became clearer (Figure 4F, Figure 7E, Figure 10G and Appendix A). This differentiation of mature bacteroids is typical for indeterminate nodules in the Inverted Repeat-Lacking Clade [24,47,48]. However, large spherical bacteroids appeared among the normal mature bacteroids in nodules induced by *R. leguminosarum* sv. *viciae* strain TOM (Figure 10B–D). These bacteroids are similar to those observed in senescent cells of *Astragalus glycyphyllos* L. [49]. However, unlike them, these bacteroids were actively released into the vacuole of the infected cell (Figure 10A,B), while the remaining bacteroids in this cell were of normal size and shape. This probably occurs as a result of fusion of the symbiosome membrane with the tonoplast (Figure 10D), which resembles exocytosis. Due to various ineffective interactions, abnormal bacteroids differentiate in nodules [50,51,52]. However, these abnormal or degenerating bacteroids are not released into the vacuole of the infected cell. Moreover, different types of bacteroids (normal and abnormal) may be present in a nodule within the same host cell [53].

In mature bacteroids in nodules induced by *R. leguminosarum* sv. *viciae* strain Vaf-108, different inclusions of varying electron density were observed, including PHB granules (Figure 7E). Previously, it was believed that in indeterminate nodules, the accumulation of PHB in bacteroids does not occur normally and is an indicator of various types of stress [36,54,55]. However, it may be a characteristic property of specific host/*Rhizobium* relationship, as was observed in nodules of *G. uralensis* [56], *Cyclopia* spp. inoculated with *Burkholderia tuberum* [57], *Anadenanthera peregrine* L. [58], and *O. maydelliana* [28]. It has been shown that *Azorhizobium caulinodans* must synthesize PHB during nitrogen fixation [59], but in pea symbioses PHB plays a less important role [54], although PHB-accumulating bacteroids are present in wild-type pea nodules [60,61]. A large-scale study of lipogenesis and redox balance in pea nodules demonstrated that bacteroids must balance NADH production from the oxidation of acetyl-coenzyme A in the citric acid cycle with its storage as PHB and lipids [62]. In this study, we observed various inclusions, other than PHB, in mature and degenerating bacteroids. As is known, inclusions with the high electron density may be nucleoids [63,64]. Inclusions with medium electron density may be either phosphates or glycogen [63,64]. The nature of other inclusions, particularly in abnormal bacteroids of the *R. leguminosarum* sv. *viciae* strain TOM (electron-transparent with a dark rim), is unknown and requires further investigation.

A striking feature of the nodules induced by *R. leguminosarum* sv. *viciae* strain TOM and *R. ruizarguesonis* strain RCAM1026 was the unusual electron- opaque and amorphous matrix, occluding the intercellular spaces in infection zone (Figure 6E and Figure 9E). Further transition to the nitrogen fixation zone of the nodule led to the appearance of translucent intercellular spaces. The occlusion of intercellular spaces in the area of early infection is associated with a differentiated oxygen barrier [65]. In addition, in nodules inoculated with *R. ruizarguesonis* strain RCAM1026, *R. leguminosarum* sv. *viciae* strains Vaf-12 and Vaf-108, electron-opaque granules of unknown nature were observed at the border with the cell wall and sealed intercellular spaces, which requires further study (Figure 2B and Appendix A).

In four-week-old *V. formosa* nodules, typical senescence zone was not observed, with the exception of the variant induced by *R. leguminosarum* sv. *viciae* strain Vaf-108. However, when inoculated with strains used for *P. sativum* inoculation, in contrast to *R. leguminosarum* sv. *viciae* strain Vaf-12 isolated from *V. formosa* nodules, individual degenerating cells appeared in the nodules (Figure 10H and Appendix A). In the case of *R. leguminosarum* sv. *viciae* strain TOM, cellular degeneration began with the degradation of bacteroids (Figure 10I), and in the case of *R. ruizarguesonis* strain RCAM1026, signs of degeneration were pyknotization of the nucleus and compaction of the cytoplasm (Appendix A).

## 4. Materials and Methods

### 4.1. Plant Material and Bacterial Strains

The strains used in this study are listed in Table 1. Their nodulation ability was confirmed in a sterile tube test experiment. It has been previously demonstrated that strains Vaf-12 and Vaf-108 form effective nodules on vavilovia plants [66]. Seeds of *V. formosa* were provided by the Gorsky State Agrarian University, Vladikavkaz, Russia. Rhizobia strains are deposited in an automated tube store (Liconic Instruments, Mauren, Lichtenstein) at the Russian Collection of Agricultural Microorganisms (RCAM, WDCM 966) in the All-Russia Research Institute for Agricultural Microbiology (ARRIAM) [67].

### 4.2. Inoculation and Plant Growth Conditions

The handling of seeds, parameters of strains cultivation and tube test conditions were described earlier [11]. Briefly, after germination, seedlings were planted in one-liter glass cylinders containing vermiculite, with N-free liquid growth medium and a solution of microelements [71] and inoculated with 1 mL of a suspension of rhizobia containing approximately 10^7^ cells. Plants were cultivated for 30 days (28 days after inoculation) in the growth chamber at 50% relative humidity with a two-level illumination/temperature mode: night (dark, 16 °C, 00 a.m.–08 a.m.), day (400 μmol m^−2^ s^−1^, 18 °C, 08 a.m.–00 a.m.). Illumination was provided by L 36W/77 Fluora lamps (Osram, Munich, Germany). Photos of nodules were taken using a Carl Zeiss Stemi 508 stereo microscope with Zeiss Axiocam ERc 5S camera (Carl Zeiss, Oberkochen, Germany).

### 4.3. Microscopy

For analyses, 15 nodules from 5 plants for each variant were harvested. Sample preparation, sectioning, contrasting for electron microscopy, and staining for light microscopy were performed as previously described [36]. Briefly, for microscopy, both light and transmission electron, nodules were transferred directly into a drop of 2.5% (*w*/*v*) glutaraldehyde diluted in 10 mM phosphate buffer (Sigma-Aldrich, St. Louis, MO, USA) at pH 7.4. For better penetration of the fixative, the cortex was cut off on one side of each nodule. The samples were then placed in a vacuum to remove air from the intercellular space.

For light microscopy, the nodules after dehydration were embedded in the London Resin White (Polysciences Europe, Eppelheim, Germany) using UV polymerization at room temperature for 48 h. The embedded material was cut into semi-thin sections (1 µm) on a Leica EM UC7 ultramicrotome (Leica Microsystems, Vienna, Austria). Sections were stained in a methylene blue-azure II solution at 70 °C for 20 min [72].

For transmission electron microscopy, the nodules were fixed overnight at 4 °C and post-fixed in a 1% aqueous solution of osmium. The nodules were then dehydrated and embedded in the EMbed-812 epoxy resin (EMS, Hatfield, PA, USA). All these procedures were performed in an EM TP tissue processor (Leica Microsystems). The samples were polymerized in an IN55 incubator (Memmert, Schwabach, Germany) at 60 °C for 48 h. Ultrathin sections (90–100 nm thick) were cut using a Leica EM UC7 ultramicrotome (Leica Microsystems). The sections were contrasted with a 2% aqueous solution of uranyl acetate for 30 min and then with lead citrate for 1 min.

The semi-thin sections were analyzed using an Axio Imager.Z1 microscope (Carl Zeiss). Photographs were taken with an Axiocam 506 digital camera (Carl Zeiss). Ultrathin sections were viewed with a JEM-1400 EM transmission electron microscope (JEOL Ltd., Tokyo, Japan) equipped with a Veleta CCD camera (Olympus, Munster, Germany).

## 5. Conclusions

This study demonstrated that both strains used for pea inoculation (RCAM1026 and TOM) and strains previously isolated from vavilovia nodules (Vaf-12 and Vaf-108) induce nodules on *V. formosa* plants. However, nodules induced by strains RCAM1026 and TOM, as well as strain Vaf-108, are characterized by various abnormalities. Only strain Vaf-12 induced typical indeterminate nodules without marked abnormalities. This indicates a pronounced specificity of the interaction of vavilovia with different strains of the *R. leguminosarum* genospecies. It may be related to the growth conditions of vavilovia in mountainous regions and the geographical isolation of its habitat. In order to confirm the observed differences in nodule structure as a regularity, it is necessary to analyze a significantly larger number of rhizobia strains. In the present study, we provide direction for further research, which is undoubtedly directly relevant to the evolution of *P. sativum* and the establishment of the evolutionary position of *V. formosa*, since vavilovia represents a unique model for studying the evolution of symbiosis in the tribe Fabeae, as it is the putative closest relative of the common ancestor for this tribe. Our study should contribute to research on the evolution of the formation of genospecies that constitute the *R. leguminosarum* complex, which is of great importance in agricultural production.

## Figures and Tables

**Figure 1 plants-14-03764-f001:**
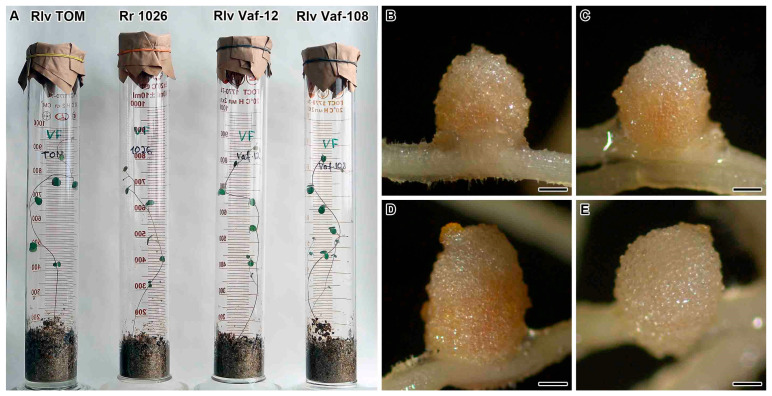
Phenotypes of *Vavilovia formosa* (Steven) Fed. 30-day-old plants (**A**) and nodules induced by *Rhizobium leguminosarum* sv. *viciae* strain TOM (**B**), *R. ruizarguesonis* strain RCAM1026 (**C**), *R. leguminosarum* sv. *viciae* strain Vaf-12 (**D**), and *R. leguminosarum* sv. *viciae* strain Vaf-108 (**E**). Nodules are shown 4 weeks after inoculation. Bars 500 µm.

**Figure 2 plants-14-03764-f002:**
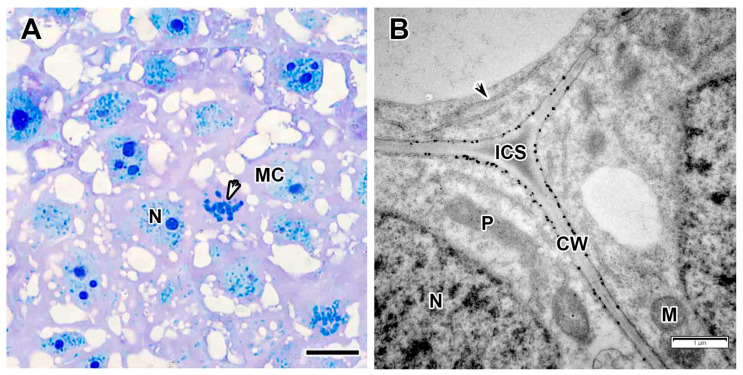
Meristem in the *Vavilovia formosa* (Steven) Fed. nodules induced by *Rhizobium leguminosarum* sv. *viciae* strain Vaf-12. (**A**) Histological organization. (**B**) Three-cell junction from the meristem. Light microscopy, methylene blue-azur II staining (**A**); transmission electron microscopy (**B**). CW, cell wall; ICS, intercellular space; M, mitochondrion; MC, meristematic cell; N, nucleus; P, proplastid; black arrowhead indicates endoplasmic reticulum profile; white arrowhead indicates mitotic figure. Bars 5 µm (**A**), 1 µm (**B**).

**Figure 3 plants-14-03764-f003:**
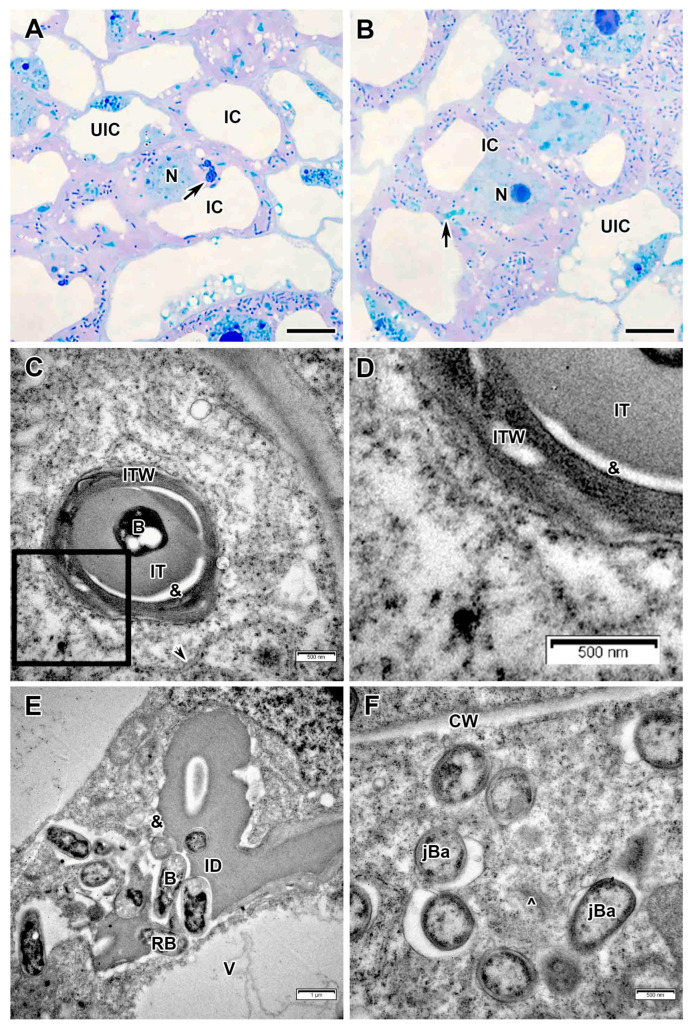
Infection zone in the *Vavilovia formosa* (Steven) Fed. nodules induced by *Rhizobium leguminosarum* sv. *viciae* strain Vaf-12. (**A**) Cells from the early infection zone. (**B**) Cells from the late infection zone. (**C**) An infection thread from the infection zone. (**D**) High magnification of the boxed area in (**C**). (**E**) An infection droplet from the infection zone. (**F**) Juvenile bacteroids in the infected cell from the infection zone. Light microscopy, methylene blue-azur II staining (**A**,**B**); transmission electron microscopy (**C**–**F**). B, bacterium; CW, cell wall; IC, infected cell; ID, infection droplet; IT, infection thread; ITW, infection thread wall; jBa, juvenile bacteroid; N, nucleus; RB, releasing bacterium; UIC, uninfected cell; V, vacuole; ^, Golgi body; &, electron-transparent ring around the infection thread matrix; black arrowheads indicate endoplasmic reticulum profiles, large black arrows indicate infection threads. Bars 5 µm (**A**,**B**), 1 µm (**E**), 500 nm (**C**,**D**,**F**).

**Figure 4 plants-14-03764-f004:**
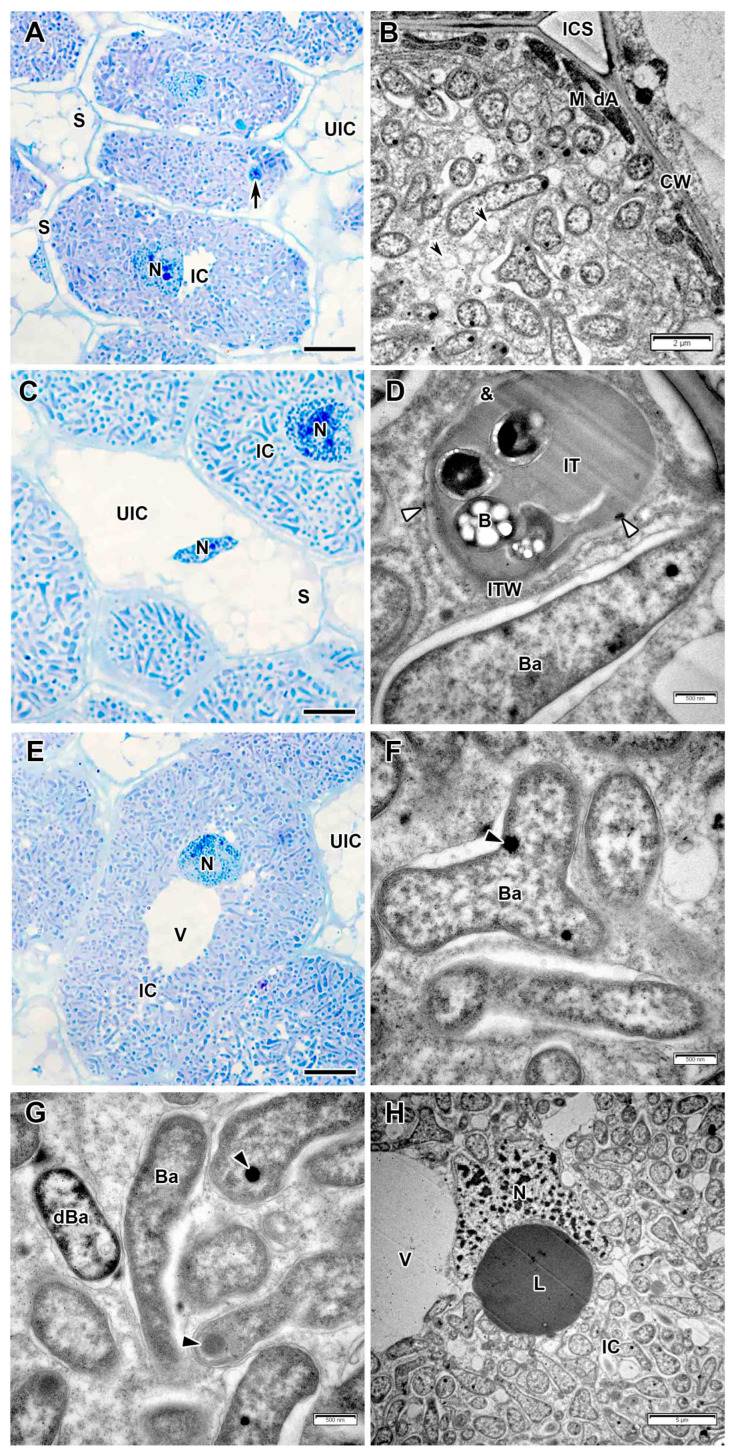
Interzone II–III and nitrogen fixation zone in the *Vavilovia formosa* (Steven) Fed. nodules induced by *Rhizobium leguminosarum* sv. *viciae* strain Vaf-12. (**A**,**B**) Cells from the interzone II–III. (**C**) An uninfected cell from the nitrogen fixation zone. (**D**) An infection thread from the nitrogen fixation zone with electron-dense accumulations around. (**E**) An infected cell from the nitrogen fixation zone. (**F**) Mature bacteroids from the nitrogen fixation zone. (**G**) A degenerating bacteroid from nitrogen fixation zone. (**H**) An infected cell from the nitrogen fixation zone with a large electron-dense, presumably lipid body. Light microscopy, methylene blue-azur II staining (**A**,**C**,**E**); transmission electron microscopy (**B**,**D**,**F**–**H**). B, bacterium; Ba, bacteroid; CW, cell wall; dA, differentiating amyloplast; dBa, degenerating bacteroid; IC, infected cell; ICS, intercellular space; IT, infection thread; ITW, infection thread wall; L, lipid body; M, mitochondrion; N, nucleus; S, starch grain; UIC, uninfected cell; V, vacuole; &, electron-transparent ring around the infection thread matrix; black arrowheads indicate expanded endoplasmic reticulum profiles; large black arrows indicate infection threads; black triangles indicate inclusions of various electron density in bacteroids; white triangles indicate electron-dense accumulations around infection thread. Bars 5 µm (**A**,**C**,**E**,**H**), 2 µm (**B**), 500 nm (**D**,**F**,**G**).

**Figure 5 plants-14-03764-f005:**
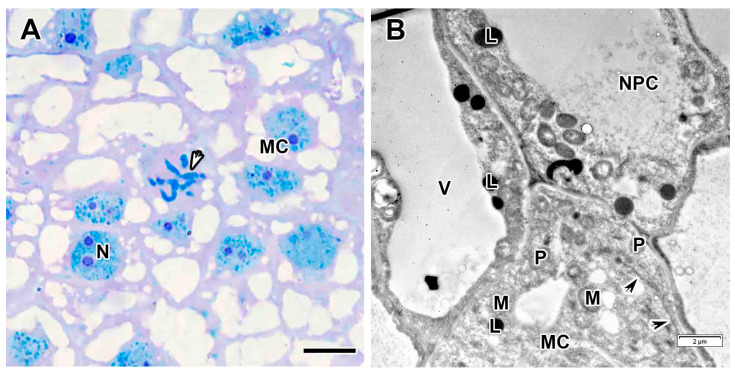
Meristem in the *Vavilovia formosa* (Steven) Fed. nodules induced by *Rhizobium leguminosarum* sv. *viciae* strain Vaf-108. (**A**) Histological organization. (**B**) Meristematic and cells of nodule parenchyma. Light microscopy, methylene blue-azur II staining (**A**); transmission electron microscopy (**B**). L, lipid body; M, mitochondrion; MC, meristematic cell; N, nucleus; NPC, nodule parenchyma cell; P, proplastid; V, vacuole; black arrowheads indicate endoplasmic reticulum profile; white arrowhead indicates mitotic figure. Bars 5 µm (**A**), 2 µm (**B**).

**Figure 6 plants-14-03764-f006:**
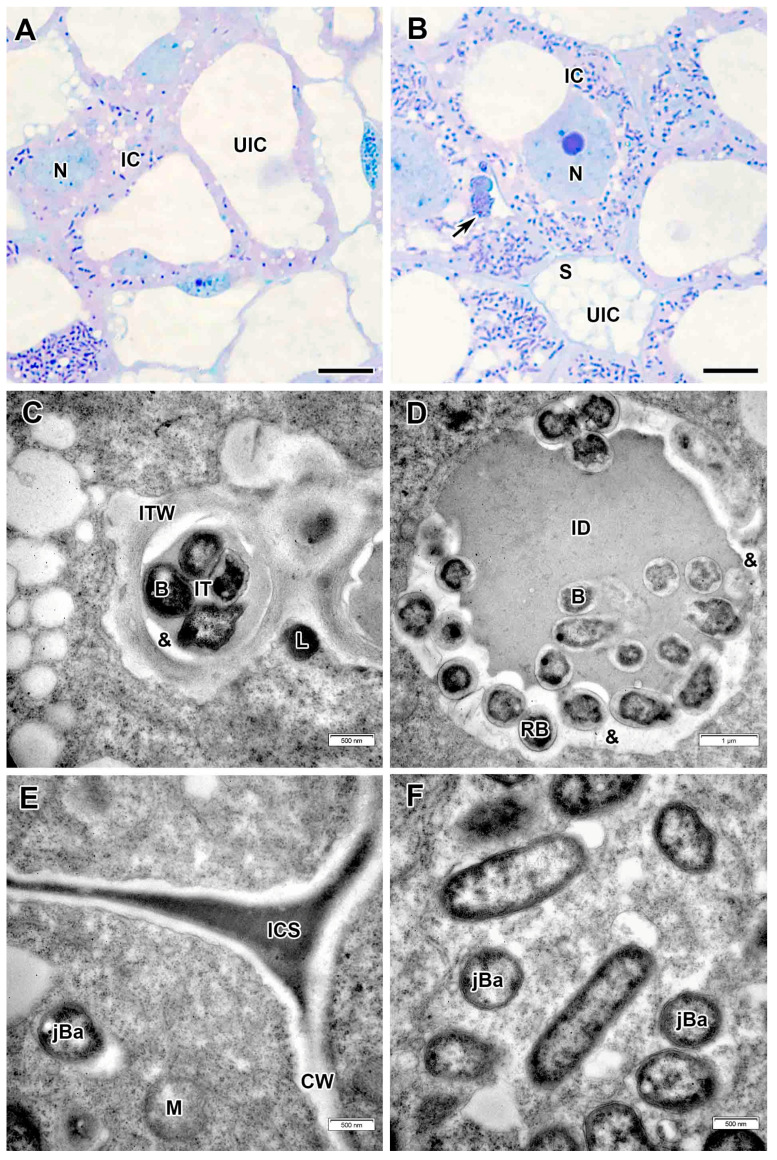
Infection zone in the *Vavilovia formosa* (Steven) Fed. nodules induced by *Rhizobium leguminosarum* sv. *viciae* strain Vaf-108. (**A**) Cells from the early infection zone. (**B**) Cells from the late infection zone. (**C**) An infection thread from the infection zone. (**D**) An infection droplet from the infection zone. (**E**) Three-cell junction from the infection zone. (**F**) Juvenile bacteroids in the infected cell from the infection zone. Light microscopy, methylene blue-azur II staining (**A**,**B**); transmission electron microscopy (**C**–**F**). B, bacterium; CW, cell wall; IC, infected cell; ICS, intercellular space; ID, infection droplet; IT, infection thread; ITW, infection thread wall; jBa, juvenile bacteroid; L, lipid body; M, mitochondrion; N, nucleus; RB, releasing bacterium; UIC, uninfected cell; &, electron-transparent ring around the infection thread and droplet matrix; large black arrow indicates infection thread. Bars 5 µm (**A**,**B**), 1 µm (**D**), 500 nm (**C**,**E**,**F**).

**Figure 7 plants-14-03764-f007:**
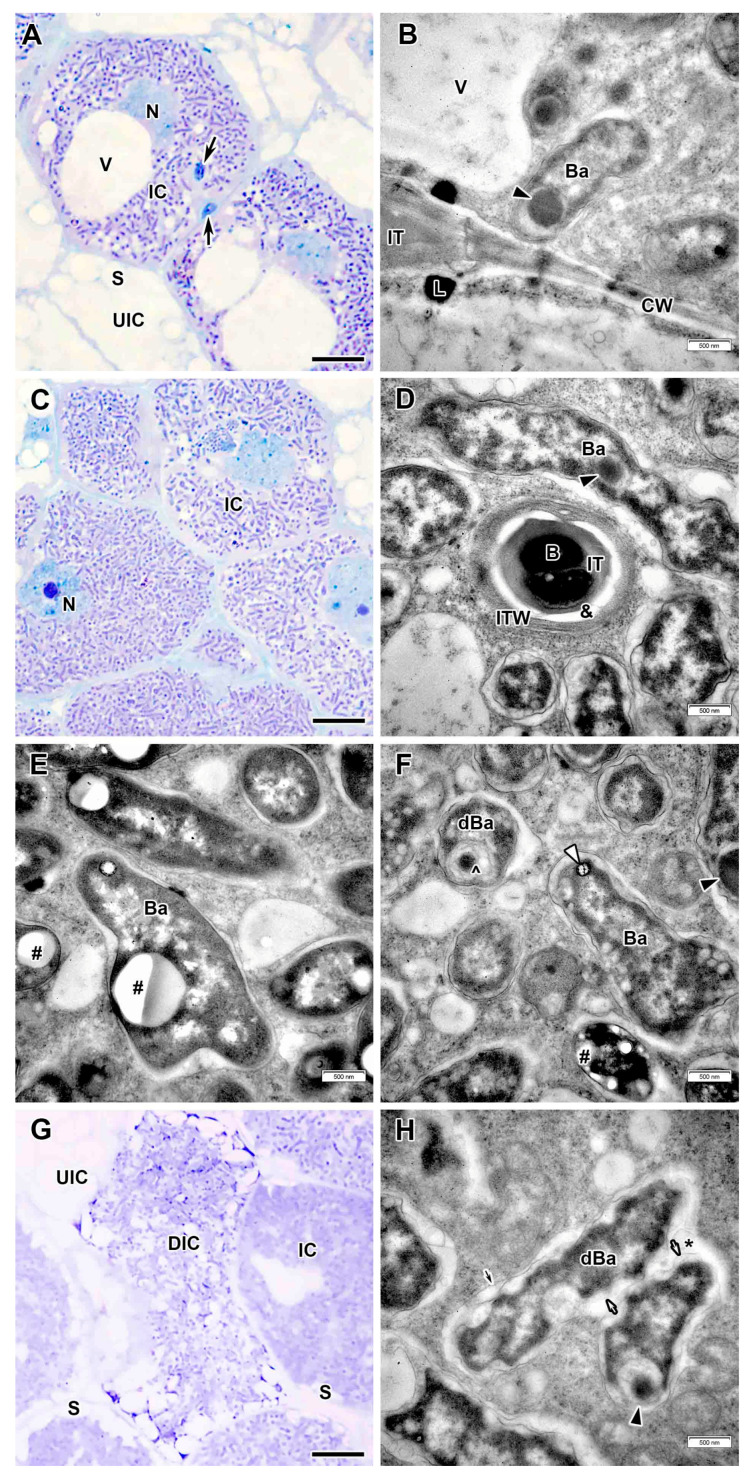
Interzone II–III and nitrogen fixation zone in the *Vavilovia formosa* (Steven) Fed. nodules induced by *Rhizobium leguminosarum* sv. *viciae* strain Vaf-108. (**A**) Cells from the interzone II–III. (**B**) An intercellular infection thread with lipid accumulations around. (**C**) Infected cells from the nitrogen fixation zone. (**D**) An infection thread from the nitrogen fixation zone. (**E**,**F**) Mature bacteroids from the nitrogen fixation zone with polyhydroxybutyrate granules (**E**), transparent inclusion with a dark ring and inclusions of various electron density (**F**). (**G**) A degrading infected cell. (**H**) Degenerating bacteroids in the degrading infected cell. Light microscopy, methylene blue-azur II staining (**A**,**C**,**G**); transmission electron microscopy (**B**,**D**–**F**,**H**). B, bacterium; Ba, bacteroid; CW, cell wall; dBa, degenerating bacteroid; DIC, degrading infected cell; IC, infected cell; IT, infection thread; ITW, infection thread wall; L, lipid body; N, nucleus; S, starch grain; UIC, uninfected cell; V, vacuole; &, electron-transparent ring around the infection thread matrix; *, multibacteroid symbiosome; #, polyhydroxibutirate granule; ^, membrane coil; black triangles indicate inclusions of various electron density in bacteroids; white triangles indicate transparent inclusion with a dark ring; large black arrows indicate infection threads; small black arrow indicates symbiosome membrane; small white arrows indicate bacteroid membrane. Bars 5 µm (**A**,**C**,**G**), 500 nm (**B**,**D**–**F**,**H**).

**Figure 8 plants-14-03764-f008:**
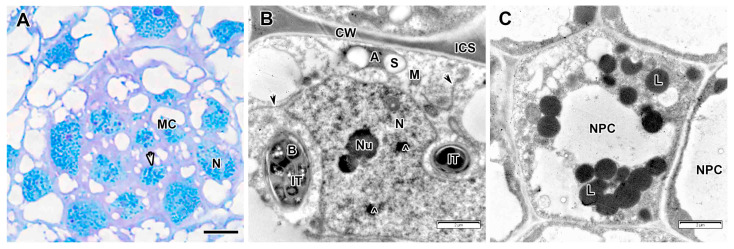
Meristem in the *Vavilovia formosa* (Steven) Fed. nodules induced by *Rhizobium leguminosarum* sv. *viciae* strain TOM. (**A**) Histological organization. (**B**) A meristematic cell. (**C**) A cell of the nodule parenchyma. Light microscopy, methylene blue-azur II staining (**A**); transmission electron microscopy (**B**,**C**). A, amyloplast, B, bacterium, CW, cell wall; ICS, intercellular space; IT, infection thread; L, lipid body; M, mitochondrion; MC, meristematic cell; N, nucleus; NPC, nodule parenchyma cell; Nu, nucleolus; S, starch grain; ^, nuclear inclusion; black arrowheads indicate endoplasmic reticulum profile; white arrowhead indicates mitotic figure. Bars 5 µm (**A**), 2 µm (**B**,**C**).

**Figure 9 plants-14-03764-f009:**
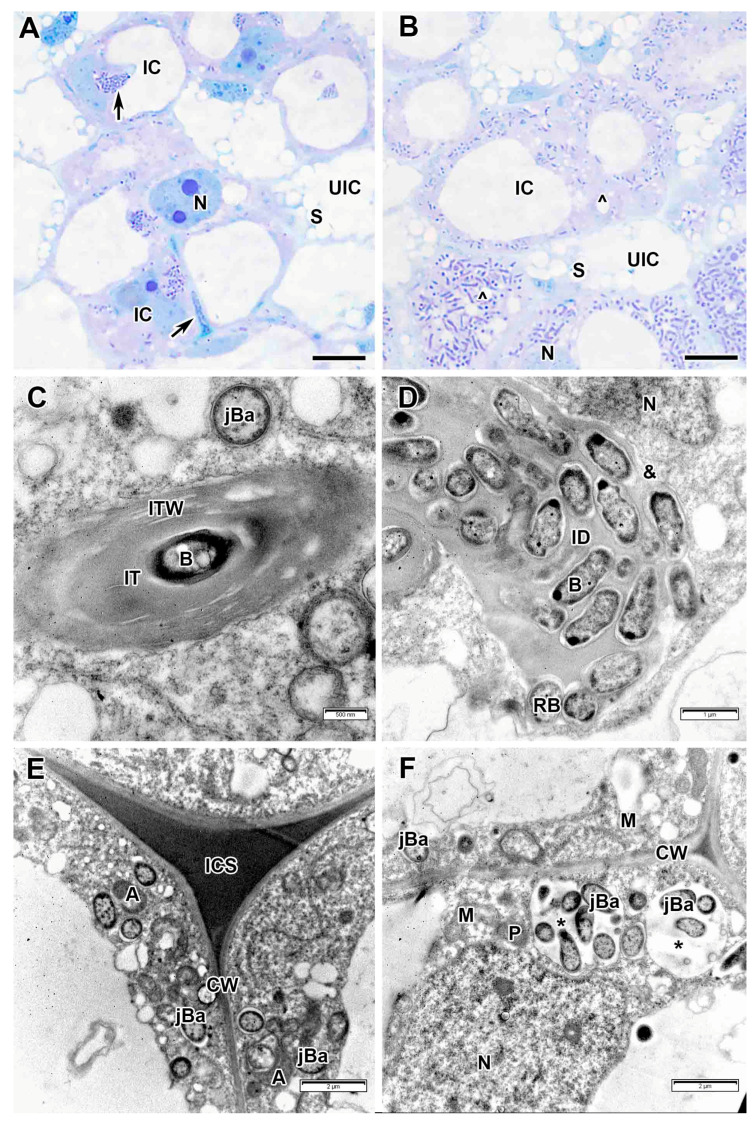
Infection zone in the *Vavilovia formosa* (Steven) Fed. nodules induced by *Rhizobium leguminosarum* sv. *viciae* strain TOM. (**A**) Cells from the early infection zone. (**B**) Cells from the late infection zone. (**C**) An infection thread from the infection zone. (**D**) An infection droplet from the infection zone. (**E**) Three-cell junction from the infection zone. (**F**) Juvenile bacteroids in the infected cell from the infection zone. Light microscopy, methylene blue-azur II staining (**A**,**B**); transmission electron microscopy (**C**–**F**). A, amyloplast; B, bacterium; CW, cell wall; IC, infected cell; ICS, intercellular space; ID, infection droplet; IT, infection thread; ITW, infection thread wall; jBa, juvenile bacteroid; M, mitochondrion; N, nucleus; P, proplastid; RB, releasing bacterium; S, starch grain; UIC, uninfected cell; &, electron-transparent ring around the infection thread matrix; *, multibacteroid symbiosome; ^, expansion of endoplasmic reticulum; large black arrows indicate infection threads. Bars 5 µm (**A**,**B**), 2 µm (**E**,**F**), 1 µm (**D**), 500 nm (**C**).

**Figure 10 plants-14-03764-f010:**
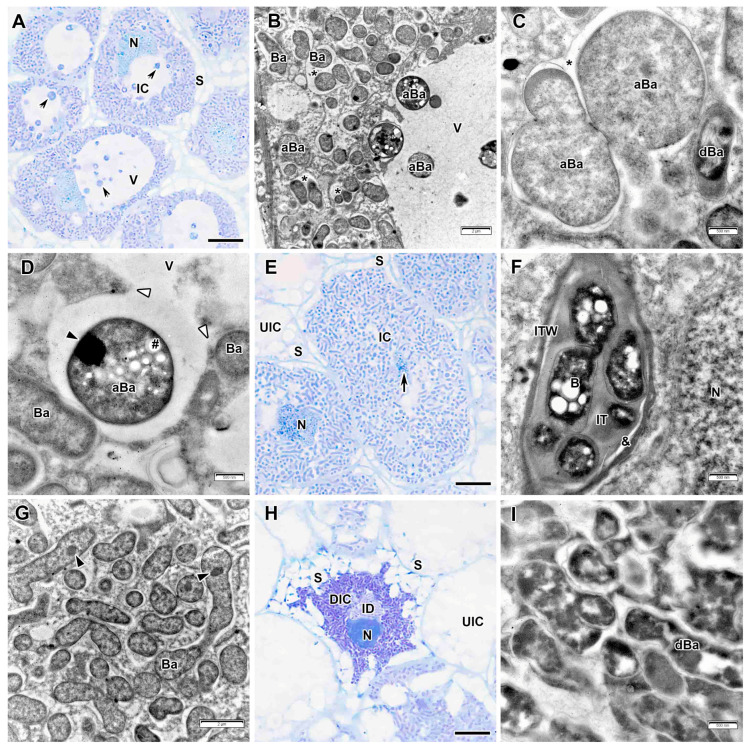
Interzone II–III and nitrogen fixation zone in the *Vavilovia formosa* (Steven) Fed. nodules induced by *Rhizobium leguminosarum* sv. *viciae* strain TOM. (**A**) Cells from the interzone II–III. (**B**–**D**) Abnormal bacteroids in the infected cells from the interzone II–III. (**C**) Initial stage of abnormal differentiation of bacteroids. (**E**) An infected cell from the nitrogen fixation zone. (**F**) An infection thread from the nitrogen fixation zone. (**G**) Mature bacteroids from the nitrogen fixation zone. (**H**) A degraded infected cell. (**I**) Degenerating bacteroids in the degrading infected cell. Light microscopy, methylene blue-azur II staining (**A**,**E**,**H**); transmission electron microscopy (**B**–**D**,**F**,**G**,**I**). aBa, abnormal bacteroid; B, bacterium; Ba, bacteroid; dBa, degenerating bacteroid; DIC, degraded infected cell; IC, infected cell; ID, infection droplet; IT, infection thread; ITW, infection thread wall; N, nucleus; S, starch grain; UIC, uninfected cell; V, vacuole; &, electron-transparent ring around the infection thread matrix; *, multibacteroid symbiosome; #, polyhydroxibutirate granule; black triangles indicate inclusions of various electron density in bacteroids; white triangles indicate a fusion of the symbiosome membrane and the tonoplast; black arrowheads indicate abnormal bacteroids in the vacuole; large black arrow indicates infection thread. Bars 5 µm (**A**,**E**,**H**), 2 µm (**B**,**G**), 500 nm (**C**,**D**,**F**,**I**).

**Table 1 plants-14-03764-t001:** Bacterial strains used in the study.

Strain, *Rhizobium* Species	Origin, Host Plant	References
Vaf-12 *Rhizobium leguminosarum* sv. *viciae*	North Ossetia, Russia, *Vavilovia formosa*	[10]
Vaf-108 *Rhizobium leguminosarum* sv. *viciae*	Dagestan, Russia, *Vavilovia formosa*	[12]
RCAM1026 (=CIAM 1026) ^1^, *Rhizobium ruizarguesonis*	Kostanay Region, Kazakhstan, *Pisum sativum*	[68,69]
TOM *Rhizobium leguminosarum* sv. *viciae*	Turkey, *Pisum sativum*, cv. Afghanistan	[70]

^1^ commercial strain.

## Data Availability

Data is contained within the article.

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
