# Peer review of "Morphology of Vavilovia formosa (Steven) Fed. Nodules Induced by Different Rhizobia Strains"

_plants, 2025, doi:10.3390/plants14243764_

Round 1
Reviewer 1 Report
Comments and Suggestions for Authors
This manuscript presents a detailed ultrastructural analysis of root nodules formed by the relict legume Vavilovia formosa when inoculated with different rhizobial strains. The study is well-designed, methodologically sound, and provides valuable insights into the symbiotic interactions between a less-studied legume and its microsymbionts. The work is of interest to the field of plant-microbe interactions, particularly in the context of legume nodulation and rhizobial specificity.
Major Concerns
- While the structural data are compelling, the study would be significantly strengthened by including nitrogen fixation measurements (e.g., acetylene reduction assay) or gene expression data (e.g., nifH) to correlate structural abnormalities with symbiotic effectiveness.
- The study is largely descriptive. Including quantitative data (e.g., percentage of abnormal bacteroids, starch granule size/number, senescence zone area) would enhance the robustness of the conclusions.
- The methods state that 15 nodules from 5 plants were used, but it is unclear how many nodules per strain were analyzed for microscopy. Statistical comparisons of observed differences are absent.
- While the micrographs are informative, some labels are small and difficult to read. Including higher-magnification insets for key abnormalities (e.g., fibrillar wall, PHB granules) would improve clarity.
Minor Points
- The manuscript is generally well-written, though some sentences are overly long or complex. Minor grammatical revisions would improve readability.
- Some parts of the results section refer to figures in a way that is hard to follow (e.g., long lists of subfigures). Streamlining this would help the reader.
- The discussion could be more focused on the ecological or evolutionary implications of the findings. For example, why might V. formosa show such strain specificity? Is this related to its perennial growth habit or geographic isolation?
- Some references in the text (e.g., [5]) are incomplete or not properly linked to the reference list.
Author Response
We thank the reviewer for the favorable review of our manuscript.
Major Concerns
While the structural data are compelling, the study would be significantly strengthened by including nitrogen fixation measurements (e.g., acetylene reduction assay) or gene expression data (e.g., nifH) to correlate structural abnormalities with symbiotic effectiveness.
This study focused specifically on the structure of nodules. Previous studies have shown that strains Vaf-12 and Vaf-108 form effective nodules on vavilovia plants.
Onishchuk O.P., Kurchak O.N., Kimeklis A.K., Aksenova T.S., Andronov E.E., Provorov N.A. Biodiversity of the symbiotic systems formed by nodule bacteria Rhizobium leguminosarum with the leguminous plants of Galegoid complex. Agricultural biology, 2023, V. 58, Iss. 1, pp. 87-99. doi: 10.15389/agrobiology.2023.1.87eng
The study is largely descriptive. Including quantitative data (e.g., percentage of abnormal bacteroids, starch granule size/number, senescence zone area) would enhance the robustness of the conclusions.
We do not believe that morphometric analysis would be useful in our study. When describing the nodules formed by the four strains, we identified qualitative differences that characterize the symbiotic pairs studied.
The methods state that 15 nodules from 5 plants were used, but it is unclear how many nodules per strain were analyzed for microscopy. Statistical comparisons of observed differences are absent.
Thanks, it was clarified.
While the micrographs are informative, some labels are small and difficult to read. Including higher-magnification insets for key abnormalities (e.g., fibrillar wall, PHB granules) would improve clarity.
We agree with the reviewer's comment. The electronic micrographs have been completely reformatted, their size has been increased.
Minor Points
The manuscript is generally well-written, though some sentences are overly long or complex. Minor grammatical revisions would improve readability.
We have tried to change the text to improve its readability.
Some parts of the results section refer to figures in a way that is hard to follow (e.g., long lists of subfigures). Streamlining this would help the reader.
Thank you for your criticism. We have taken your comment into account and reduced the number of photos in one plate and the figure legend.
The discussion could be more focused on the ecological or evolutionary implications of the findings. For example, why might V. formosa show such strain specificity? Is this related to its perennial growth habit or geographic isolation?
While we do not have enough data for a detailed discussion, we have added this assumption to the conclusion.
Some references in the text (e.g., [5]) are incomplete or not properly linked to the reference list.
The list of references was checked.
Reviewer 2 Report
Comments and Suggestions for Authors
The paper presents detailed findings about the structure of nodules in Vavilovia formosa, a legume species, inoculated with different rhizobial strains. It can be overly complex for a broader audience, especially for those not well-versed in plant microbiology. The use of intricate terms like "pleiomorphic bacteroids" and "electron-opaque matrix" without clear explanation could make the content inaccessible to non-specialists.
The paper could benefit from a clearer connection between the experimental findings and their real-world applications.
In results, there is a lack of synthesis between the individual findings, making it difficult for the reader to understand their broader implications. In addition, the presentation of the results could benefit from a more structured comparison between strains and a clearer connection to the study's objectives.
The authors should discuss the limitations of the study, particularly regarding the scope of rhizobial strains tested and the potential for variability in other environmental conditions.
The discussion section delves into technical details about nodule morphology, but the broader implications of the findings are not adequately explored.
The figures should explain the relevance of these observations and why they matter, especially in terms of how they relate to the broader field of legume biology and agricultural applications. Without this context, the figures feel like disconnected pieces of information, making it harder for readers to fully grasp the significance of the research.
The manuscript offers valuable insights into the morphology of Vavilovia formosa nodules and the role of rhizobia, but it could benefit from a more concise presentation of results, clearer explanations of technical terms, and a stronger connection to broader agricultural applications.
Author Response
We thank the reviewer for reading our manuscript.
The paper presents detailed findings about the structure of nodules in Vavilovia formosa, a legume species, inoculated with different rhizobial strains. It can be overly complex for a broader audience, especially for those not well-versed in plant microbiology. The use of intricate terms like "pleiomorphic bacteroids" and "electron-opaque matrix" without clear explanation could make the content inaccessible to non-specialists.
We have used terms that are widely used in scientific literature to describe ultrastructure; interested readers will undoubtedly be able to find their meanings whenever necessary.
The paper could benefit from a clearer connection between the experimental findings and their real-world applications.
Our research is primarily basic, and its practical application may emerge in the future. However, we have made some speculations in conclusion.
In results, there is a lack of synthesis between the individual findings, making it difficult for the reader to understand their broader implications. In addition, the presentation of the results could benefit from a more structured comparison between strains and a clearer connection to the study's objectives.
In our opinion, we have compared the studied strains in sufficient detail and identified both common features of nodule formation and strain-specific features.
The authors should discuss the limitations of the study, particularly regarding the scope of rhizobial strains tested and the potential for variability in other environmental conditions.
We see no particular need to draw the reader's attention to the limitations of our study, as they are obvious. The materials and methods section provides a detailed description of the plant growth conditions and strains used.
The discussion section delves into technical details about nodule morphology, but the broader implications of the findings are not adequately explored.
We appreciate your suggestion, but we tried to be more specific in the discussion, discussing the results obtained and avoiding overly general speculation.
The figures should explain the relevance of these observations and why they matter, especially in terms of how they relate to the broader field of legume biology and agricultural applications. Without this context, the figures feel like disconnected pieces of information, making it harder for readers to fully grasp the significance of the research.
We are confident that specialists will understand our figures, as we have seen from the feedback of other reviewers. Nevertheless, the figures have been modified for better understanding.
The manuscript offers valuable insights into the morphology of Vavilovia formosa nodules and the role of rhizobia, but it could benefit from a more concise presentation of results, clearer explanations of technical terms, and a stronger connection to broader agricultural applications.
Reviewer 3 Report
Comments and Suggestions for Authors
Dear Authors,
Recently, I had a privilege to review your manuscript “Morphology of Vavilovia formosa (Steven) Fed. Nodules Formed by Different Rhizobia Strains”.
The manuscript provides microscopic analysis of root nodules formed by little known host plant in symbiosis with various rhizobial strains. Therefore, it must be published, and I emphasize this with full conviction. It seems to me that the manuscript's particular strength is its valuable TEM documentation - provided that some images reduced in size in the manuscript in a way that makes analysis of their content impossible will be published as much larger ones.
However, due to some problems listed below, the manuscript must be substantially improved.
The minor issues are indicated in the manuscript annotated file (which is an integral part of this review) and I will not discuss them here. They are mostly of editorial nature, plus some technical problems, some icorrect/not optimal uses of terms or minor misinterpretations of structure.
The most important problems require re-thinking by the Authors and must be adequately corrected. They include the following:
1. The title does ot correspond to the manuscript content. "Morphology" refers to the external appearance. In this study, the Authors analyzed the internal structure of the nodules, not their morphology. Another problem is, the Authors use consistently a synonymous latin name of the host plant studied (Vavilovia). As the Authors themselves mention, this genus was included in Lathyrus sp. The correct name is not even mentioned in the manuscript.
A similar problem occurs in the subchapter titles and figure captions (ultrastrcture of the nodules) in the Results – they do not correspond to the content, as only the structure of the bacteroid tissue is presented and the other nodule tissues are omitted.
2. Introduction
The host plant is little known, in comparison with related genera. Therefore I recommend adding brief information on the species: e.g., perennial/annual, typical habitat, herbaceous/woody, vine or not.
Since the Introduction has to provide a background on the problem studied, adding a brief synopsis on the indeterminate nodule anatomy/ultrastructure is necessary.
3. Materials and methods
Important data are lacking. Even if the methodology was the same as in previous papers (duly cited by Authors), the basic information must be included. The specific requirements are listed in the annotated manuscript file. Here just the most important ones: it must be stated, which rhizobia are the commercial strains and which strain(s) can be considered (or expected) to be fully compatible with the host plant studied. Also, it must be stated clearly, at what age of plants the nodules were sampled (days after inoculation) and how many nodules from a single plant were taken - or better, how many nodules per plant were actually sectioned and analyzed. This would show whether the Authors obtained data that sufficiently reflected the genetic variability within the host species. Since no commercial cultivar was used, the variability may be high. Next: what constituted the replication in test tube or in microscopic analysis?
4. Results
Figure 2 (the histological structure of the nodules studied, light microscope images)
These images are technically poor and not acceptable, because important anatomical features are not discernible or they are lacking:
4.1. The cell walls are not distinguishable, especially in the cortical tissues, making them completely impossible to analyze.
4.2. The cell nuclei (and the cells themselves) are not distinguishable in the meristematic zone in Fig. 2A and D, making it impossible to assess whether the meristem is actually present in the sections shown – and this is very important, because only the presence of actually dividing cells determines whether the meristem is present or not.
Moreover, no dividing cells are shown in the (supposed) meristems in all four images (and in the TEM images). It may indicate a problem that is often encountered with test-tube obtained plants. Namely, it is very often in tube tests, that the nodules lose the mitotic activity early (a few weeks after inoculation) and cease growth due to the suboptimal root conditions (nutrients, space, sometimes some exudates that in high concentrations are toxic) limiting the "fitness" of the host and its ability to "feed" the nodules. The initial stage of the meristem activity cessation is not so spectacular and the structure of the nodule apex is then easy to misunderstand. The arrest of mitoses, at least for some time, is reversible (if optimal growth conditions are restored), and during this period we can speak of a latent meristem. However, if no mitoses are found, this must be clearly stated in the Results and the reason(s) for this raised in the Discussion.
As a side note: Since vavilovia is a perennial, it's a pity the Authors did not cultivate the plants in a way that ensured their long-term growth and undisturbed nodule development. Hopefully, in the future, the Authors will allow us to understand the structure of nodules that develop over many months. It will be interesting to see, e.g., if they branch in time, if this year's nodules are shed during the plant's dormancy period, or if they continue to develop (and then, how are they prepared to the dormancy).
4.3. The connection with the root is lacking in B-D, and this part of the nodule is no less important than the bacteroid tissue. Correct functioning of the long distance transport is one of the determinants of the nodule efficiency, although it is rarely investigated. Nodules incapable of exporting N2-fixation products have no reason to exist, from host's perspective. Note how rarely viable mutants with defective long-distance transport are obtained.
Overall, the images look as if they were taken with a poorly adjusted microscope, specifically with the aperture diaphragm too wide open. Perhaps it would be best to capture the images anew.
4.4. The anatomy of the nodules is not properly labeled, in fact, only the bacteroid-containing parenchymatic tissue is labelled. Labels indicating nodule endodermis, inner and outer cortex and vascular bundles must be added, also the II/III interzone should be indicated (if present). It would be best to add images (as inserts?) showing 1) the dividing cells in the meristems, 2) the boundary layer (if present) in the inner cortex, 3) the vascular bundle (are the transfer cells present?).
The lack of labels for root nodule tissues other than bacteroid tissue causes the figure caption not to correspond to the figure content.
Figure 3 and the next TEM images:
All figures documenting ultrastructure follow an identical pattern: all images are squeezed into a regular square array, regardless of the image magnification or its content. This is fundamentally incorrect, as it sacrifices content for form. The arrangement of images in a regular array is neither important nor valuable. What is fundamentally important is whether a particular image actually documents the results of the analysis. Therefore, all images must be published at a size that clearly shows all the features that the individual images are intended to document. For example, all images showing entire cells or large cell fragments (i.e., the first 3-4 images in every figure) should be published at least twice as large as proposed, because in their current state they document nothing — they are so small that little can actually be seen (as opposed to: imagined). Note that enlarging these images in the pdf file does not help much because instead of showing more detail, they become pixelated and blurry.
It must be observed that in structural studies (anatomy, histology, ultrastructure), what is not documented by an image is not treated as fact but as anecdote because other researchers are unable to evaluate this information.
Other issues:
# in no point it was proven that the small vacuoles are in fact expanded ER cisterns, it is just supposition and as such, it can be placed in Discussion but not Results;
# some details described in text are not discernible in images, e.g., the fibrillar structure of the infection thread wall - again the matter of magnification;
# the presence of the II/III interzone is completely omitted, and this cell layer is very typical for the bacteroid tissue of indeterminate nodules;
# the ultrastructure of the uninfected cells of the bacteroid tissue is mostly omitted;
# sometimes the interpretation of the ultrastructure is questionable, e.g. the presence of an infection thread clearly indicates that the cell has entered the differentiation, so is therefore no longer a meristematic cell;
# an important ultrastructural feature of the differentiated infected cells is location of amyloplasts and the majority of mitochondria near the intercellular spaces; unfortunately, most of the presented images (implicitly: the representative ones) do not show the cell corners - it is impossible to assess whether the symbioses studied in the manuscript are similar or not in terms of this feature.
4. Discussion
Overally, no major problems, but I recommend that, after the first sentence of this chapter, the Authors begin the Discussion by specifying which combination of the host and rhizobial strain are to be considered fully compatible, thus providing a basis for all comparisons. It should be clear what is typical of vavilovia nodules and what constitutes an aberration. At the present form of the manuscript, such statement is only in the... abstract ;-)
I hope the above helps.
With best regards,
Sincerely yours,
reviewer

Author Response
We are extremely grateful to the reviewer for their high appreciation of our work, as well as for their detailed analysis of the article and valuable comments, which we have tried to take into account in the new version.
The manuscript provides microscopic analysis of root nodules formed by little known host plant in symbiosis with various rhizobial strains. Therefore, it must be published, and I emphasize this with full conviction. It seems to me that the manuscript's particular strength is its valuable TEM documentation - provided that some images reduced in size in the manuscript in a way that makes analysis of their content impossible will be published as much larger ones.
Thank you for your feedback. We have modified the figures and enlarged the images.
However, due to some problems listed below, the manuscript must be substantially improved.
The minor issues are indicated in the manuscript annotated file (which is an integral part of this review) and I will not discuss them here. They are mostly of editorial nature, plus some technical problems, some icorrect/not optimal uses of terms or minor misinterpretations of structure.
We thank the reviewer for providing a file with comments, which greatly contributed to improving the manuscript.
The most important problems require re-thinking by the Authors and must be adequately corrected. They include the following:
- The title does ot correspond to the manuscript content. "Morphology" refers to the external appearance. In this study, the Authors analyzed the internal structure of the nodules, not their morphology. Another problem is, the Authors use consistently a synonymous latin name of the host plant studied (Vavilovia). As the Authors themselves mention, this genus was included in Lathyrus sp. The correct name is not even mentioned in the manuscript.
With all due respect to the reviewer, we cannot agree with the use of the term morphology, as confirmed by the definition given in the encyclopedia.
The Encyclopædia Britannica: Plant morphology studies plant structure at a range of scales. At the smallest scales are ultrastructure, the general structural features of cells visible only with the aid of an electron microscope, and cytology, the study of cells using optical microscopy. At this scale, plant morphology overlaps with plant anatomy as a field of study.
The currently accepted name for vavilovia was used in the manuscript.
A similar problem occurs in the subchapter titles and figure captions (ultrastrcture of the nodules) in the Results – they do not correspond to the content, as only the structure of the bacteroid tissue is presented and the other nodule tissues are omitted.
We do not see a major problem with using the subheading “Nodule Structure” in the title, as we are considering the structure of various zones of the nodule, including the meristem, the infection zone, and the nitrogen fixation zone. These zones are of greatest interest when studying the structure of the nodule, since the structure of the peripheral tissues is similar in nodules formed on plants belonging to the same genus. The term “bacteroid tissue” is undoubtedly outdated. In those figures where only the ultrastructure of infected cells is shown, we have added clarifications to the figure titles.
- Introduction
The host plant is little known, in comparison with related genera. Therefore I recommend adding brief information on the species: e.g., perennial/annual, typical habitat, herbaceous/woody, vine or not.
The information about Vavilovia formosa species has been published by our research team earlier
Safronova V.I., Kimeklis A.K., Chizhevskaya E.P., Belimov A.A., Andronov E.E., Pinaev A.G., Pukhaev A.R., Popov K.P., Tikhonovich I.A. Genetic diversity of rhizobia isolated from nodules of the relic species Vavilovia formosa (Stev.) Fed.// Antonie van Leeuwenhoek, 2014, V.105, p.389–399. DOI 10.1007/s10482-013-0089-9
A brief description has also been added to the introduction.
Since the Introduction has to provide a background on the problem studied, adding a brief synopsis on the indeterminate nodule anatomy/ultrastructure is necessary.
We understand the reviewer's suggestion, but we wanted to avoid another description of the process of forming a indeterminate nodule in this manuscript, which has been done many times by us and other researchers in numerous articles. In our opinion, interested readers can always refer to these publications. Nevertheless, we have moved the brief description of the structure of indeterminate nodules from the Discussion to the Introduction.
- Materials and methods
Important data are lacking. Even if the methodology was the same as in previous papers (duly cited by Authors), the basic information must be included. The specific requirements are listed in the annotated manuscript file. Here just the most important ones: it must be stated, which rhizobia are the commercial strains and which strain(s) can be considered (or expected) to be fully compatible with the host plant studied. Also, it must be stated clearly, at what age of plants the nodules were sampled (days after inoculation) and how many nodules from a single plant were taken - or better, how many nodules per plant were actually sectioned and analyzed. This would show whether the Authors obtained data that sufficiently reflected the genetic variability within the host species. Since no commercial cultivar was used, the variability may be high. Next: what constituted the replication in test tube or in microscopic analysis?
We have tried to provide explanations for all the questions asked by the reviewer.
- Results
Figure 2 (the histological structure of the nodules studied, light microscope images)
These images are technically poor and not acceptable, because important anatomical features are not discernible or they are lacking:
4.1. The cell walls are not distinguishable, especially in the cortical tissues, making them completely impossible to analyze.
4.2. The cell nuclei (and the cells themselves) are not distinguishable in the meristematic zone in Fig. 2A and D, making it impossible to assess whether the meristem is actually present in the sections shown – and this is very important, because only the presence of actually dividing cells determines whether the meristem is present or not.
Thank you for your feedback. New photographs have been received, including high-resolution and high-magnification images. The manuscript includes photographs of cells from different areas of the nodule, showing cell walls, nuclei, and bacteroids.
Moreover, no dividing cells are shown in the (supposed) meristems in all four images (and in the TEM images). It may indicate a problem that is often encountered with test-tube obtained plants. Namely, it is very often in tube tests, that the nodules lose the mitotic activity early (a few weeks after inoculation) and cease growth due to the suboptimal root conditions (nutrients, space, sometimes some exudates that in high concentrations are toxic) limiting the "fitness" of the host and its ability to "feed" the nodules. The initial stage of the meristem activity cessation is not so spectacular and the structure of the nodule apex is then easy to misunderstand. The arrest of mitoses, at least for some time, is reversible (if optimal growth conditions are restored), and during this period we can speak of a latent meristem. However, if no mitoses are found, this must be clearly stated in the Results and the reason(s) for this raised in the Discussion.
The electron micrographs were replaced with light microscopy images, where the mitotic figures were clearly visible.
As a side note: Since vavilovia is a perennial, it's a pity the Authors did not cultivate the plants in a way that ensured their long-term growth and undisturbed nodule development. Hopefully, in the future, the Authors will allow us to understand the structure of nodules that develop over many months. It will be interesting to see, e.g., if they branch in time, if this year's nodules are shed during the plant's dormancy period, or if they continue to develop (and then, how are they prepared to the dormancy).
Oh, how we tried to grow vavilovia! Since it grows in high-altitude conditions, it simply does not develop properly in laboratory conditions. As a result, we were unable to keep it alive for more than one season — it died every winter. And, of course, we couldn't get it to bloom and produce seeds, so our efforts were limited to very small seed stocks.
4.3. The connection with the root is lacking in B-D, and this part of the nodule is no less important than the bacteroid tissue. Correct functioning of the long distance transport is one of the determinants of the nodule efficiency, although it is rarely investigated. Nodules incapable of exporting N2-fixation products have no reason to exist, from host's perspective. Note how rarely viable mutants with defective long-distance transport are obtained.
Overall, the images look as if they were taken with a poorly adjusted microscope, specifically with the aperture diaphragm too wide open. Perhaps it would be best to capture the images anew.
The photos were changed.
4.4. The anatomy of the nodules is not properly labeled, in fact, only the bacteroid-containing parenchymatic tissue is labelled. Labels indicating nodule endodermis, inner and outer cortex and vascular bundles must be added, also the II/III interzone should be indicated (if present). It would be best to add images (as inserts?) showing 1) the dividing cells in the meristems, 2) the boundary layer (if present) in the inner cortex, 3) the vascular bundle (are the transfer cells present?).
The lack of labels for root nodule tissues other than bacteroid tissue causes the figure caption not to correspond to the figure content.
The photos were changed.
Figure 3 and the next TEM images:
All figures documenting ultrastructure follow an identical pattern: all images are squeezed into a regular square array, regardless of the image magnification or its content. This is fundamentally incorrect, as it sacrifices content for form. The arrangement of images in a regular array is neither important nor valuable. What is fundamentally important is whether a particular image actually documents the results of the analysis. Therefore, all images must be published at a size that clearly shows all the features that the individual images are intended to document. For example, all images showing entire cells or large cell fragments (i.e., the first 3-4 images in every figure) should be published at least twice as large as proposed, because in their current state they document nothing — they are so small that little can actually be seen (as opposed to: imagined). Note that enlarging these images in the pdf file does not help much because instead of showing more detail, they become pixelated and blurry.
It must be observed that in structural studies (anatomy, histology, ultrastructure), what is not documented by an image is not treated as fact but as anecdote because other researchers are unable to evaluate this information.
Your comments have been taken into account. Overview photos of the cells were taken using light microscopy, and details of the cells were captured using electron microscopy. In addition, the images were enlarged for better visibility.
Other issues:
# in no point it was proven that the small vacuoles are in fact expanded ER cisterns, it is just supposition and as such, it can be placed in Discussion but not Results;
We have described this in the results as a hypothesis, since these extensions are currently observed under light microscopy.
# some details described in text are not discernible in images, e.g., the fibrillar structure of the infection thread wall - again the matter of magnification;
It was corrected.
# the presence of the II/III interzone is completely omitted, and this cell layer is very typical for the bacteroid tissue of indeterminate nodules;
It was corrected.
# the ultrastructure of the uninfected cells of the bacteroid tissue is mostly omitted;
It was corrected.
# sometimes the interpretation of the ultrastructure is questionable, e.g. the presence of an infection thread clearly indicates that the cell has entered the differentiation, so is therefore no longer a meristematic cell;
We understand the reviewer's doubts, but we believe that Figure 9B shows precisely such an example. Furthermore, it should be noted that vavilovia belongs to the same genus as pea, for which the penetration of infection threads into dividing meristem cells has been demonstrated.
# an important ultrastructural feature of the differentiated infected cells is location of amyloplasts and the majority of mitochondria near the intercellular spaces; unfortunately, most of the presented images (implicitly: the representative ones) do not show the cell corners - it is impossible to assess whether the symbioses studied in the manuscript are similar or not in terms of this feature.
It was done.
- Discussion
Overally, no major problems, but I recommend that, after the first sentence of this chapter, the Authors begin the Discussion by specifying which combination of the host and rhizobial strain are to be considered fully compatible, thus providing a basis for all comparisons. It should be clear what is typical of vavilovia nodules and what constitutes an aberration. At the present form of the manuscript, such statement is only in the... abstract ;-)
Thank you, we tried to use your suggestion.
Reviewer 4 Report
Comments and Suggestions for Authors
The manuscript presents a significant contribution to the field of crop legumes. The results are presented clearly. However, the experimental design is not robust, and a few improvements in terms of clarity, consistency, and presentation are recommended to enhance the overall readability and impact of the paper.
Why did the author just isolate several nodule bacteria? And re-inoculation is needed to show the effectiveness.
Why didn’t the author assay the nitrogenase activity? Nodule morphology is good, but it is not sure it can fix nitrogen.
In the materials and methods, the description of the plant growth and treatment application section is clear.
The discussion makes good use of the results but could be enhanced by more clearly linking the findings with broader agricultural practices.
The conclusion is well-written, but it would be good to propose specific future research directions based on the current findings.
Author Response
We thank the reviewer for their kind feedback and positive appreciation of our research.
Why did the author just isolate several nodule bacteria? And re-inoculation is needed to show the effectiveness.
Strains Vaf-12 and Vaf-108 were previously isolated from vavilovia nodules and used in previous experiments to study symbiotic efficiency (see references 9-11 in the MS). Strains CIAM1026 and TOM have a history of research as pea inoculants spanning several decades.
Why didn’t the author assay the nitrogenase activity? Nodule morphology is good, but it is not sure it can fix nitrogen.
This study focused specifically on the structure of nodules. Previous studies have shown that strains Vaf-12 and Vaf-108 form effective nodules on vavilovia plants.
Onishchuk O.P., Kurchak O.N., Kimeklis A.K., Aksenova T.S., Andronov E.E., Provorov N.A. Biodiversity of the symbiotic systems formed by nodule bacteria Rhizobium leguminosarum with the leguminous plants of Galegoid complex. Agricultural biology, 2023, V. 58, Iss. 1, pp. 87-99. doi: 10.15389/agrobiology.2023.1.87eng
The discussion makes good use of the results but could be enhanced by more clearly linking the findings with broader agricultural practices.
The conclusion is well-written, but it would be good to propose specific future research directions based on the current findings.
We focused on fundamental issues, but tried to mention the possible practical application of the results in the conclusion.
Round 2
Reviewer 2 Report
Comments and Suggestions for Authors
Thank you for the authors’ responses. While I appreciate the revisions made, several replies remain problematic and do not meet standard expectations for scientific publishing. In particular, some responses dismiss essential aspects of manuscript clarity and scientific rigor. I would like to emphasize the following:
The authors state that readers may “find their meanings whenever necessary,” but this is not acceptable.
Scientific manuscripts must define specialized or uncommon terms at first use to ensure clarity for all readers, including researchers from adjacent fields.
All scientific studies require an explicit limitations or constraints section. It is not sufficient to assume readers will infer them. The manuscript must clearly discuss:
-
the limited number of strains tested
-
environmental conditions
-
controlled growth settings
-
potential variability in natural systems
This is essential for transparency and for guiding future research.
Figures should be interpretable by any informed reader of the journal, not just specialists in ultrastructure. Each figure should include enough explanatory context to clarify why the observation matters and how it ties into the study’s objectives.
The earlier comment requested a clearer synthesis of findings and a short explanation of broader relevance to legume biology. This was not a request for speculation, but for contextualization. The current response does not address that need.
Several key issues remain unresolved, not because of experimental limitations but because of the authors’ reluctance to revise. I request that the authors fully address the points above to ensure the manuscript meets the clarity, transparency, and accessibility standards expected in scientific publishing.
Author Response
We thank the reviewer for the second round of review.
Thank you for the authors’ responses. While I appreciate the revisions made, several replies remain problematic and do not meet standard expectations for scientific publishing. In particular, some responses dismiss essential aspects of manuscript clarity and scientific rigor. I would like to emphasize the following:
The authors state that readers may “find their meanings whenever necessary,” but this is not acceptable.
Scientific manuscripts must define specialized or uncommon terms at first use to ensure clarity for all readers, including researchers from adjacent fields.
We understand the need to explain the terms used in articles. However, the challenge is determining which terms should be defined and which should not. We followed reviewer’s kind advice and added a paragraph to the introduction explaining the key terms used in the article: infection thread, infection droplet, bacteroid, symbiosome, peribacteroid (symbiosome) membrane, and peribacteroid space (lines 60-73).
All scientific studies require an explicit limitations or constraints section. It is not sufficient to assume readers will infer them. The manuscript must clearly discuss:
the limited number of strains tested
environmental conditions
controlled growth settings
potential variability in natural systems
This is essential for transparency and for guiding future research.
We cannot agree with some of the reviewer's statements. Our article clearly demonstrates that the use even four strains (with a description of the differences between these strains) allowed us to identify differences in nodule phenotypes. Moreover, it is quite enough to demonstrate the polymorphism in the nodule structures. In the conclusion we added mention of direction of further studies, which lies in the evolutionary area (but they are not in scope of this manuscript). We can consider this manuscript as a part of the bigger study we are working at.
The Materials and Methods section describes the plant growth conditions under which the analyzed nodule phenotypes were obtained. All discussions in the article are based on the description of nodules obtained under these conditions. It is obvious that any changes in plant growth conditions (including growing them under uncontrolled conditions) can lead to the different phenotypic manifestations. It seems trivial to point this out.
Figures should be interpretable by any informed reader of the journal, not just specialists in ultrastructure. Each figure should include enough explanatory context to clarify why the observation matters and how it ties into the study’s objectives.
In current modified version, figures contain all the necessary information for understanding them: the objects of study, the microscopy techniques used, the zones or cells of the nodule visible in the photograph, and all symbiotic and cellular structures are indicated. Numerous abnormalities are highlighted.
The earlier comment requested a clearer synthesis of findings and a short explanation of broader relevance to legume biology. This was not a request for speculation, but for contextualization. The current response does not address that need.
In revised conclusion, we presented the significance of our data for the biology of legumes (lines 565-571).
Reviewer 3 Report
Comments and Suggestions for Authors
Dear Authors,
I appreciate the corrections that were made to the manuscript. I especially value the new LM documentation - it truly facilitates better understanding of the nodules under analysis. And it is very pretty, additionally :-)
However, from the perspective of the reader, it would be much better if the images were arranged in figures according to the developmental zones of the nodules induced by a particular strain (meristem, infection zone, nitrogen fixation zone, etc.) rather than the microscopic technique (light microscopy, TEM). This would then allow for the good practice of maintaining the same order of image references in the text and their placement in figures. Currently, the references to LM and TEM images are mixed and we have sort of chaos (e.g., at the beginning of Results it is 2A, 3A, 2CD, 2BF, when it should be 2A, 2B, 2C, and the following text is no better) that makes reading of the manuscript and checking the descriptions against the documentation quite a task ;-) . I strongly recommend to re-arrange the images into combined LM-TEM figures documenting the nodule zones - as they are described in the text. Seems like much work but it is technical one.
Also, as new images and text were added, new issues have arisen concerning some use of terminology, of image labels etc. - please check comments in the annotated manuscript file. I am afraid that after the third round of annotations and changes, the text may be difficult to decrypt ;-)
Additionally, check the following paper on Fabeae taxonomy - apparently it is still under discussion, so perhaps it should be mentioned in the Introduction (Ellis, T.H.N.; Smýkal, P.; Maxted, N.; Coyne, C.J.; Domoney, C.; Burstin, J.; Bouchenak-Khelladi, Y.; Chayut, N. The taxonomic status of genera within the Fabeae (Vicieae),
with a special focus on Pisum. Diversity 2024, 16, 365. https://doi.org/10.3390/d16070365).
I sympathize with the problems of long-term cultivation of vavilovia, I understand them perfectly, because I once came across similar ones. Still, you were able to produce quite extensive documentation of high quality :-) I hope you find the time and resources to continue your research, and vavilovia will eventually give in ;-)
With best regards,
Sincerely yours,
reviewer

Author Response
We are extremely grateful to the Reviewer for the repeated detailed analysis of our article.
However, from the perspective of the reader, it would be much better if the images were arranged in figures according to the developmental zones of the nodules induced by a particular strain (meristem, infection zone, nitrogen fixation zone, etc.) rather than the microscopic technique (light microscopy, TEM). This would then allow for the good practice of maintaining the same order of image references in the text and their placement in figures. Currently, the references to LM and TEM images are mixed and we have sort of chaos (e.g., at the beginning of Results it is 2A, 3A, 2CD, 2BF, when it should be 2A, 2B, 2C, and the following text is no better) that makes reading of the manuscript and checking the descriptions against the documentation quite a task ;-) . I strongly recommend to re-arrange the images into combined LM-TEM figures documenting the nodule zones - as they are described in the text. Seems like much work but it is technical one.
We took this reviewer's comment into account and reformatted the figures.
Also, as new images and text were added, new issues have arisen concerning some use of terminology, of image labels etc. - please check comments in the annotated manuscript file. I am afraid that after the third round of annotations and changes, the text may be difficult to decrypt ;-)
We understand the difficulties in reviewing the revised text and apologize for this. We have included both the clean copy and the copy with the highlighted changes. Unfortunately, the MDPI Publisher only sends the revised version to reviewers.
Additionally, check the following paper on Fabeae taxonomy - apparently it is still under discussion, so perhaps it should be mentioned in the Introduction (Ellis, T.H.N.; Smýkal, P.; Maxted, N.; Coyne, C.J.; Domoney, C.; Burstin, J.; Bouchenak-Khelladi, Y.; Chayut, N. The taxonomic status of genera within the Fabeae (Vicieae), with a special focus on Pisum. Diversity 2024, 16, 365. https://doi.org/10.3390/d16070365).
We are grateful for this reference and used it as a justification for retaining the traditional names of the species, although we also indicated the proposed "new" names.
Below we provide responses to your comments in the text of the manuscript.
In Materials and methods methylene blue/azure is mentioned, but here, it seems to me that metachromasy typical of toluidine blue is visible rather than bluish/greenish tones typical of Meth/Az method.
the nodules in LM images of the 1st version of the manuscript were colored typically for methylene blue/azure - note how different is the coloration in this figure.
It should be noted that the photographs presented in the current version of the manuscript were taken using LRW resin-embedded sections of nodules. In the first version, the sections were embedded in Epon resin. Below, we present two photographs illustrating the differences in coloration observed when using Meth/Az and TolBlue stains on sections embedded in LRW resin.
Figure 10A: incorrectly labeled - this is a bacteroid with inclusion.
Thanks for this comment. Indeed, it is not an amyloplast; upon closer inspection, we found out it was a mitochondrion, not a bacteroid. However, this micrograph was removed during subsequent modification of the figure.
Figure 10B: jBa? rather bacteria in the infection droplet (note the relatively dense matrix around these rhizobia), which was cut so that its connection with the inf thread is not in the section plane.
This micrograph became Figure 9F when the figures were rearranged. We are sure that this is symbisome, because in infection structures such as infection threads and infection droplets, the matrix has a more electron-dense and sometimes granular or fibrillar structure. In this case, the electron density of the matrix surrounding the juvenile bacteroids corresponds to the density of the peribacteroid space.
arrow: a concave plastid?
Yes, it is possible, but it was not labeled.
parenchymatous nodule inner cortex
With the exception of vascular tissues and meristem, all other nodule tissues are specialized parenchymas (if we take into account the classical division of plant tissues), therefore it is reasonable to use names that describe the location in the nodule or the main function of the individual tissues.
We use the terms and definitions given in the comprehensive review Guinel (2009): “the terms ‘‘outer cortex’’ and ‘‘nodule parenchyma’’ will be used for indeterminate nodules”.
it would be better to refer to symbiosomes rather than lone bacteroids - after all, the symbiosome is a functional unit that undergoes differentiation as a whole
We fully agree with you that symbiosomes undergo differentiation in infected cells. This process involves changes in the molecular markers of symbiosome membranes, in addition to changes in the bacteroids. However, ultrastructural studies that do not utilize various symbiosome markers visually detect differentiation of the bacteroids, not the symbiosome membranes. Therefore, in this manuscript and similar publications, the term "bacteroids" is used to describe the differentiation process.
Is there any literature available on the ultrastructure of pea nodules induced by the pea rhizobia strains used in the present study? It would be interesting to discuss the (putative) ultrastructural differences of the rhizobia cells persisting in the infection threads and bacteroids differentiating in the root nodules of pea (the fully-compatible host) and vavilovia.
Unfortunately, to our knowledge, the structure of the pea nodules it induced is known only for strain 1026. The pea nodules differed to some extent from those of vavilovia. We have added a note of this to the discussion.
- polyhydroxybutyrate granules atypically large and few - is this typical for vavilowia symbionts? add comment
Unfortunately, we cannot confirm that such large polyhydroxybutyrate granules are characteristic of vavilovia symbionts. We are currently unable to compare them, as we have only one example of granules of this size observed. Perhaps such studies will be conducted in the future, and then it will be possible to determine with certainty whether large PHB granules are a characteristic feature of vavilovia symbionts.
- Cajal bodies signify high activity, it should be commented in Discussion
Indeed, the appearance of Cajal bodies is due to high functional activity, which could certainly be related to meristematic activity, as Cajal bodies have been detected in cancer cells. However, we will not discuss this in the article, as we are not completely certain that these are indeed Cajal bodies, as we did not detect coilin.
- interestingly, the bacteroids (or whole symbiosomes? clarify please) included into the vacuole contain PHB already in contrast to these in the normal symbiosomes
We believe that it is the bacteroid, and not the entire symbiosome that is release into the vacuole, as can be seen in Figure 10D. The clarification was given in Discussion.
- highest? not so: nucleoid is electron dense but not opaque, and looks like a bundled fibre, relatively thick in comparison to bacterial cell sizes
the polyphosphate-containing inclusions have an electron opaque portion - they can look like heterogenous globules.
Also, lipid or proteinaceous globules can occure, and they are moderately electron dense, and if material fixation is perfect the proteinaceous ones are finely granular at high magnifications.
Generally, difficult matter, as reliable comparative documentation is scarce.
We are grateful to the reviewer for providing a wide range of literature on bacterial inclusions. Indeed, data (micrographs) on inclusions in our study are sparse, as they were not the subject of our investigation. Our interpretation relied on electron microscopic data obtained on rhizobia (publications) and can only speculate about the nature of these inclusions, as we did not conduct additional studies.

Reviewer 4 Report
Comments and Suggestions for Authors
Thank you for the revision, as well as some reasonable arguments. The manuscript can be accepted now.
Author Response
Dear Reviewer,
We sincerely thank you for your positive feedback on our article.
Round 3
Reviewer 2 Report
Comments and Suggestions for Authors
The revised and responses are upto mark.